# Lantern-shaped flexible RNA origami for *Smad4* mRNA delivery and growth suppression of colorectal cancer

Muren Hu[1,2,4], Chang Feng[3,4], Qianqin Yuan[1,4], Chenbin Liu[1], Bujun Ge ◉[2]✉, Fenyong Sun ◉[1]✉ & Xiaoli Zhu ◉[1]✉

mRNA delivery has shown high application value in the treatment of various diseases, but its effective delivery is still a major challenge at present. Herein, we propose a lantern-shaped flexible RNA origami for mRNA delivery. The origami is composed of a target mRNA scaffold and only two customized RGD-modified circular RNA staples, which can compress the mRNA into nanoscale and facilitate its endocytosis by cells. In parallel, the flexible structure of the lantern-shaped origami allows large regions of the mRNA to be exposed and translated, exhibiting a good balance between endocytosis and translation efficiency. The application of lantern-shaped flexible RNA origami in the context of the tumor suppressor gene, *Smad4* in colorectal cancer models demonstrates promising potential for accurate manipulation of protein levels in in vitro and in vivo settings. This flexible origami strategy provides a competitive delivery method for mRNA-based therapies.

Colorectal cancer (CRC), one of the main contributors to cancer-associated mortality globally, is characterized by the gradual changes in the genetic and epigenetic profile of the normal colonic epithelium[1,2]. Mutation of the suppressor of mothers against decapentaplegic homolog 4 (*Smad4*) gene is widely considered to be associated with the progression of CRC, and restoration of its expression can suppress CRC cell proliferation and metastasis[3–7]. Generally, manipulation of protein expression is one of the common approaches in genetic disease therapy. However, directly delivering protein is hard to achieve due to the characteristics of the protein[8]. Recently, mRNA has received extensive attention in tumor therapy as an alternative to protein-based drugs. As the direct template of protein synthesis, mRNA can be translated into protein without entering the nucleus, avoiding the risk of gene contamination of host cells[9,10]. Moreover, it is accessible to design and synthesize the mRNA of almost any protein in vitro, which makes mRNA-based therapy a perspective treatment strategy for various diseases[11,12]. However, several limitations such as the instability of mRNA and hard to be uptaken by cells

restrain its therapeutic application[13]. Approaches of encapsulation of target mRNA into nanoparticles have been taken to overcome these obstacles including the complexation of mRNA with cationic polymers, peptides, or lipids[14–17]. Nevertheless, the unwanted immune responses and cytotoxicity of these vectors induced by chemical components are still inevitable[18–21]. Therefore, exploring the low-toxic and efficient mRNA delivery system is urgently needed.

The essence of the current delivery strategy is to wrap macromolecular linear mRNA into nano-sized carriers to obtain better stability for delivery. Therefore, the nanolization of macromolecular mRNA molecules is the key to efficient delivery. DNA origami technology that can fold a long single-stranded DNA or RNA into a controllable nanostructure by hybridizing with a series of staple sequences through complementary base pairing realizes the nanolization process of macromolecular long-chain nucleic acid[22]. Origami structure has better physicochemical properties than its linear structure for delivery and storage[23,24]. Extensive literature has shown that DNA origami can be delivered into cells and perform their

[1]Department of Clinical Laboratory Medicine, Shanghai Tenth People's Hospital, School of Medicine, Tongji University, Shanghai 200072, P. R. China. [2]Department of General Surgery, Tongji Hospital, School of Medicine, Tongji University, Shanghai 200065, P. R. China. [3]Center for Molecular Recognition and Biosensing, School of Life Sciences, Shanghai University, Shanghai 200444, P. R. China. [4]These authors contributed equally: Muren Hu, Chang Feng, Qianqin Yuan. ✉e-mail: gebujun@126.com; sunfenyong@263.net; xiaolizhu@shu.edu.cn

corresponding functions[25,26]. However, it is a pity that the nanostructures formed by traditional DNA origami technology are fully double-stranded compact structures[27], which theoretically can be used for mRNA nanolization, but cannot release mRNA for protein expression effectively. From this perspective, a strategy that can both nanolize and loosen mRNA may be the key to realizing origami-based mRNA delivery.

In this work, we develop a lantern-shaped flexible origami for efficient delivery of *Smad4* mRNA, which acts as both a gene-drug cargo and a scaffold of the origami. Unlike conventional origami, which uses hundreds of linear oligonucleotides as staples, only two customized circular RNAs are adopted here to staple a few key sites of mRNA intermittently, making the mRNA form a three-dimensional lantern-shaped nanostructure and leaving most of the region of mRNA single-stranded and active in parallel. Moreover, we modify the RNA staple sequence with an RGD peptide that can specifically bind to overexpressed integrin receptors on the membrane of colorectal cancer cells[28] to endow the lantern-shaped nanostructure with the targeting capability. After uptake by cells, mRNA in the lantern-shaped nanostructure can be recognized by ribosomes in the cytoplasm. With the initiation of protein translation mediated by ribosomes, mRNA can be dissociated from nanostructures competitively and express the coded protein efficiently, which presents a suppression effect on CRC both in vitro and in vivo. In this way, we develop a flexible origami strategy to enable origami-based nanolization and translation of mRNA, and thus achieve facile and biocompatible mRNA delivery.

## Results

### Principle of the lantern-shaped flexible origami for Smad4 mRNA delivery

As shown in Fig. 1a, *Smad4* mRNA was synthesized through the transcription method (IVT) method and then capped with an m7G cap at its 5′-terminal and tailed with a poly (A) tail at its 3′-terminal, which would contribute to maintaining the stability of mRNA and promoting its translation ability. Liner RNA staples with 5′-terminal phosphorylation were self-linked into circular staple RNAs (CS-RNAs) through T4 RNA Ligase. To realize the target delivery ability of the mRNA nano-lantern, RGD (Arg-Gly-Asp), a widely used biocompatible small-molecule targeting peptide, is conjugated with the circular RNA staple through Sulfo-SMCC reaction (Fig. 1b). To assemble the compressed nano-

lantern structure of mRNA, two circular staple RNAs (CS RNAs) were adopted to anchor the *Smad4* mRNA through the complementary base pairing of several spaced binding sites (Fig. 1c). Because of the restriction of the circular RNA staples, the mRNA is expected to surround the CS RNA scaffold with a certain degree of flexibility but in a confined nanospace. The nano-lantern structure may lie between two theoretical extremes (Fig. 1d): one is the exposed strands of the mRNA are fully stretched to form a slender "lantern" (Right); the other is the circular RNA staples are close together and the exposed strands of the mRNA extend sideways, forming a depressed "lantern" (Left). In either case, the structure of the nano-lantern always lies in the nanoscale in three dimensions with a maximum of no more than 90.8 nm ideally for *Smad4* mRNA delivery. The circular staple RNAs here provide confinement of the target mRNA, and in parallel can resist the degradation of nucleases, thereby facilitating the stabilization of the mRNA nanostructures during in vivo transport. While interacting with cells, as shown in Fig. 1e, the mRNA nano-lantern could be uptaken by the target cells through its nanostructure as well as the interaction between the RGD and the overexpressed integrin receptor on the cell membrane. With the start of the translation process meditated by the m7G cap, the mRNA could be recognized by the intracellular ribosome so that it can separate from the nano-lantern and translate into protein due to its flexible structure. Subsequently, the overexpression of Smad4 protein mediated by the mRNA nano-lantern could suppress the expression of its downstream oncogenes and play a negative role in the growth and metastasis progression of colorectal cancer.

### Construction and characterization of the mRNA nano-lantern

Electrophoresis was first employed to characterize the synthesis of the *Smad4* mRNA, the RGD-CS RNA, and the mRNA nano-lantern. As shown in Fig. 2a, *Smad4* mRNA (lane 3) was synthesized from the linear template plasmid DNA containing a T7 promoter (lane 2 & Supplementary Fig. 1) through an IVT method and then successfully added with an m7G cap at its 5′-terminal and a poly (A) tail at its 3′-terminal respectively with a larger molecular weight (lane 4 & 5) in agarose electrophoresis. The formation of 17 nt dsRNA was reported could enhance the tolerability of mRNA against nuclease attack without great influence on its translation ability[29,30]. Therefore, as for the synthesis of CS RNA, shown in Fig. 2b, two artificially predetermined liner RNAs (lane 2 and lane 6), each of which contains five equally spaced 17 bp

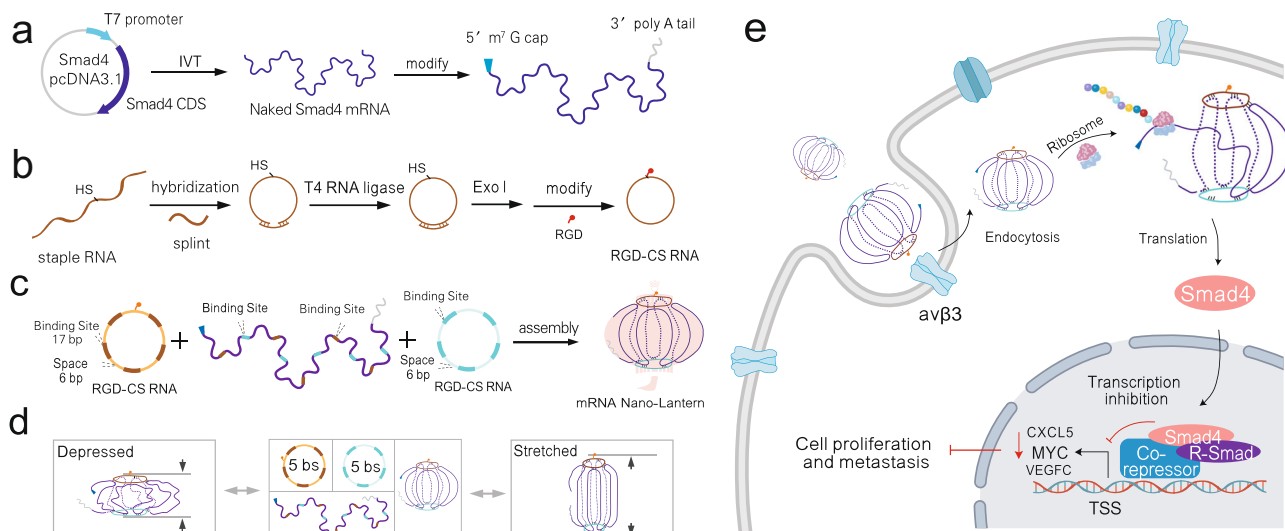

**Fig. 1 | Schematic illustration of *Smad4* mRNA nano-lantern. a** Synthesis and modification of *Smad4* mRNA. **b** Synthesis of RGD-modified circular staple RNA. **c** Construction of lantern-shaped flexible origami. **d** Possible structures of the

*Smad4* mRNA nano-lantern (5 bs). **e** Delivery of the mRNA nano-lantern into targeted cells and potential use in studying growth suppression in colorectal cancer. Created with BioRender.com.

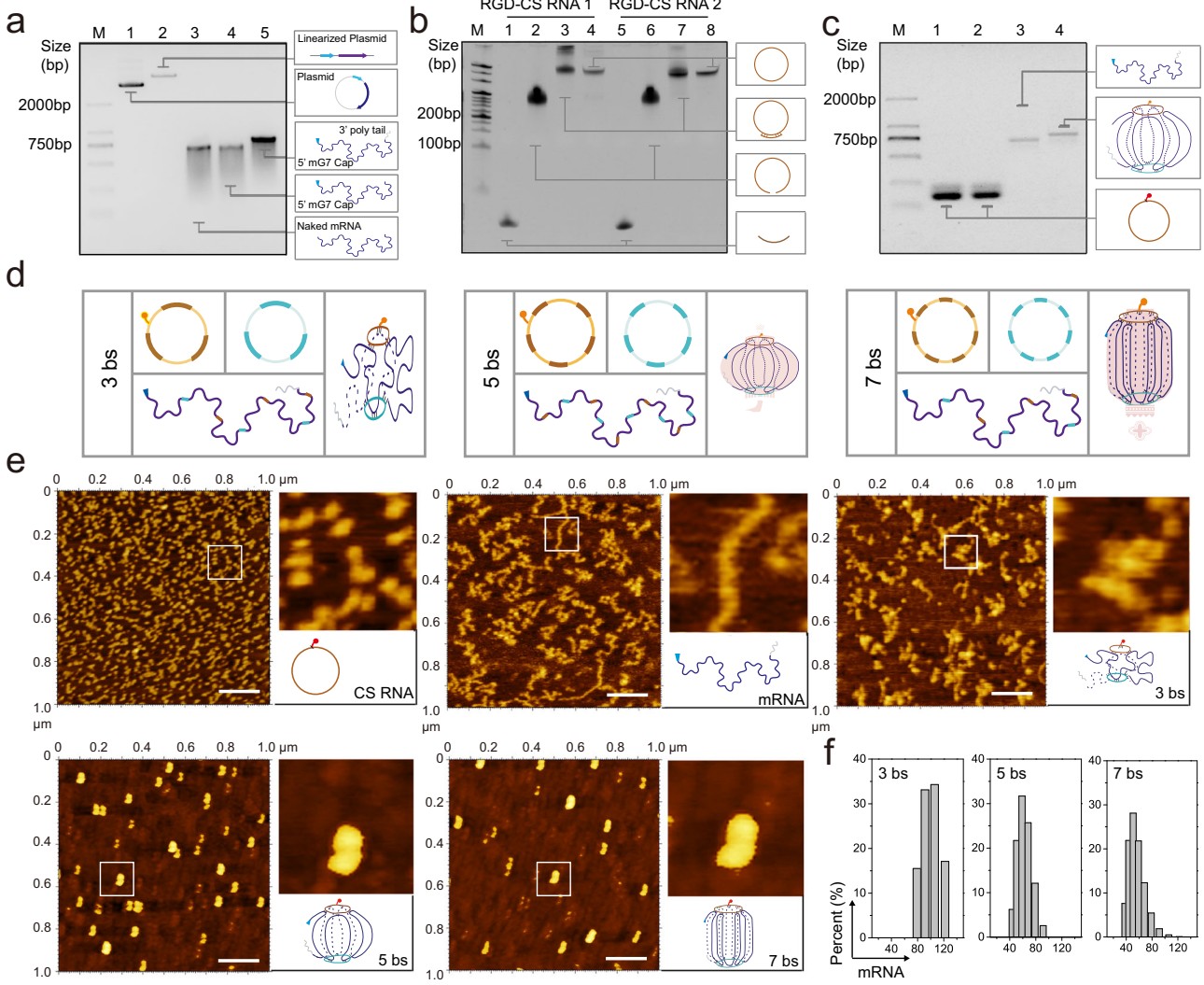

**Fig. 2 | Construction and characterization of *Smad4* mRNA nano-lantern.**
**a**−**c** Representative images from 3 independent experiments. Agarose gel electrophoresis of mRNA synthesis and modified (**a**), urea-polyacrylamide gel electrophoresis of circular RNA staples (5 bs) synthesize (**b**), and formation of *Smad4* mRNA nano-lantern (**c**). (**d**) Possible structures of the *Smad4* mRNA nano-lantern with different binding sites. (**e**) Representative AFM imaging of naked *Smad4* mRNA and different forms of mRNA nano-lantern (with 3, 5, and 7 binding sites) from 3 independent experiments, Scale bar: 0.2 μm. (**f**) Diameter of different forms of mRNA nano-lantern. Panel (**c**−**e**) created with BioRender.com. Source data are provided as a Source Data file.

length mRNA binding regions, were adopted as the raw materials. After ligation and degradation by exonuclease, only cyclized RNA can be preserved (lane 4 and lane 8), suggesting the successful synthesis of the CS RNA couple. Then, the RGD was modified on the CS RNA through a Sulfo-SMCC reaction (Supplementary Fig. 2). To explore the effect of different binding sites on translation, in addition to the CS RNA couple with 5-binding sites (5 bs), CS RNA couples with 3- or 7-binding sites were also synthesized (Fig. 2c, d, Supplementary Fig.3, 4). In the case of the *Smad4* mRNA nano-lantern, compared to individual mRNA, there is an observable shift of the position of the bands while the *Smad4* mRNA scaffold bound two corresponding CS RNA with either 3-, 5- or 7-binding sites (Fig. 2c and Supplementary Fig.3, 4). These results suggest that a complex of the *Smad4* mRNA with the staple couple is successfully formed.

Atomic force microscopy (AFM) was further employed to characterize the morphology of the mRNA nano-lantern as well as the individual mRNA and RGD-CS RNA (Fig. 2e and Supplementary Fig.5). The RGD-CS RNA with varying numbers of binding regions all displayed a uniform dot appearance under AFM. Linear *Smad4* mRNA otherwise exhibited an expected curly linear shape under AFM

(Fig. 1e). After being stapled to the RGD-CS RNA, the *Smad4* mRNA showed a condensed appearance (Fig. 2e). It is interesting that as the number of binding sites increased, the degree of compression of the nano-lantern became larger, and the average size decreased, respectively (Fig. 2e, f). In addition, the regularity of the structure increases with the binding sites, presenting a loose and irregular structure in the case of 3 bs and a relatively dense and regular structure in which the two dot-shaped CS-RNAs get close to each other due to the bridging of mRNA in the case of 5 and 7 bs. Dynamic light scattering (DLS) analysis also supported the results of AFM by showing shrinkage sizes with the increased number of binding sites (Fig. 2f).

To test the stability of the mRNA nano-lantern, it was incubated with 10% fetal bovine serum (FBS) for 2, 4, and 12 h, respectively. Through agarose gel electrophoretic analysis (Fig. 3a, b), it is found that the free mRNA was degraded rapidly within 2 h, while the nano-lanterns were preserved in the first few hours and were degraded slowly, showing observable bands even after 12 h of incubation. We also found that an increase in the number of binding sites can make the mRNA more stable (mRNA « 3 bs < 5 bs ≈ 7 bs). Combined with the AFM results, we could conclude that the results are expected and consistent

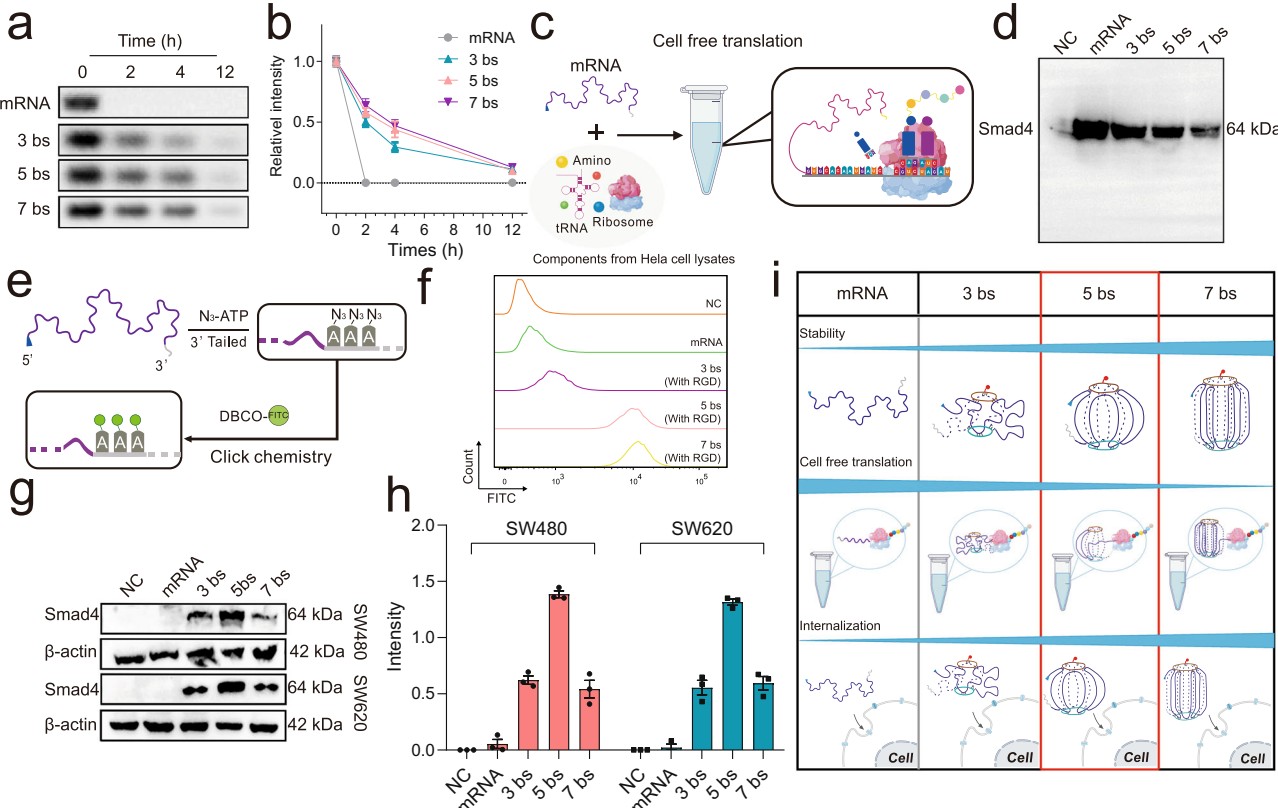

**Fig. 3 | Screening of nano-lanterns with different binding sites. a** Agarose gel electrophoresis of stability of naked mRNA and different forms of mRNA nano-lantern treated with 10% fetal bovine serum (FBS) for 2, 4, and 12 h, respectively. **b** Intensity statistical of panel (**a**), $n = 3$ independent experiments, data are presented as the mean ± SEM. **c** Schematic illustration of cell-free translation of mRNA. (**d**) Western blot analysis of protein expression level of individual mRNA and different forms of mRNA nano-lantern (3, 5, and 7 bs) in the cell-free translation system, $n = 3$ independent experiments. **e** Schematic illustration of the construction of FITC labeled mRNA. (**f**) Intake profiles of FITC labeled nano-lantern (with 3, 5, and 7 bs) in SW480 cell. The gating strategy was provided in the Supplementary Fig. 16a. **g** Representative western blot analysis of protein expression level in SW480 and SW620 cells treated with individual mRNA and different forms of mRNA nano-lantern (3, 5, and 7 bs) from 3 independent experiments. **h** Intensity statistical of panel (**g**), $n = 3$ independent biological samples, data are presented as the mean ± SEM. **i** Schematic illustration of screening of nano-lanterns with different binding sites. Panel (**c**, **e**, **i**) created with BioRender.com. Source data are provided as a Source Data file.

with some literature that the compressed structure of origami contributes to the stability of nucleic acids[29,31].

Next, we explored the intracellular delivery efficiency and the translation activity of the mRNA nano-lantern. To explore the influence of nanolization on mRNA translation ability, we used the cell-free translation system to detect the in vitro translation efficiency of different nanostructures (3, 5, and 7 bs) (Fig. 3c). As shown in the Fig. 3d, the translation ability of naked mRNA is the strongest. With the increase of binding sites, the protein in vitro translation efficiency of mRNA gradually decreased. Which indicated the excess binding sites will impede translation.

To facilitate the in situ observation of mRNA delivery, azido-modified poly A tail was used instead of ATP during the process of mRNA tailing, followed by click chemistry labeling with a fluorophore (DBCO-FITC) (Fig. 3e). Flow cytometry analysis revealed that RGD helps the enhancement of mRNA nano-lantern transfection, especially for 5 bs and 7 bs structure (Supplementary Fig. 6). In addition, the cellular uptake efficiency of nanostructures with 5 and 7 bs was much higher than that of 3 bs and the individual mRNA group (Fig. 3f and Supplementary Fig.6) which because the nano-lanterns with 5 and 7 bs are compacted enough to be endocytosed, while the one with 3 bs is still too loose to be endocytosed effectively. For the translation activity in cells, as shown in Fig. 3g, we incubated SW480 and SW620 cells, two typical Smad4-null colorectal cancer cell lines[4,32], with the nano-lanterns for Western blot analysis. Results showed that RGD-modified

nano-lantern could improve the intracellular yield of Smad4 protein (Supplementary Fig.7). The 5 bs nano-lantern group exhibited the highest Smad4 protein expression level compared with other groups, suggesting its successful protein translation ability in cells (Fig. 3h). Different from the results in the cell-free translation system, the protein expression of the 3 bs group was not significantly improved in cells, which may be limited by the low intracellular delivery efficiency (Fig. 3f). As for the 7 bs group, although it has good delivery efficiency (Fig. 3f), its protein expression was still unsatisfactory. It may be due to the too compacted structure, which is not easy to dissociate and translate within the cell. Combined with the results above, the RGD-modified nano-lantern with 5 bs was the optimal choice (Fig. 3i), which can balance delivery efficiency and translational activity.

## Delivery and translation of the mRNA nano-lantern in vitro

We next evaluated the dynamic uptake process of the delivery of 5 bs mRNA nano-lantern into CRC cells through flow cytometry and laser-scanning confocal microscope imaging. As shown in Fig. 4a, after 1 h incubation, about 72.2% of SW480 cells could swallow the nano-lantern, and almost all cells (98.8%) could ingest the nano-lantern after extending the incubation time to 2 h, which was further set as the optimum incubation time.

We then explored the cellular uptake mechanism of mRNA nano-lantern in SW480 cells by using different endocytic inhibitors. The inhibitors are as follows: wortmannin (WTM, inhibitor of

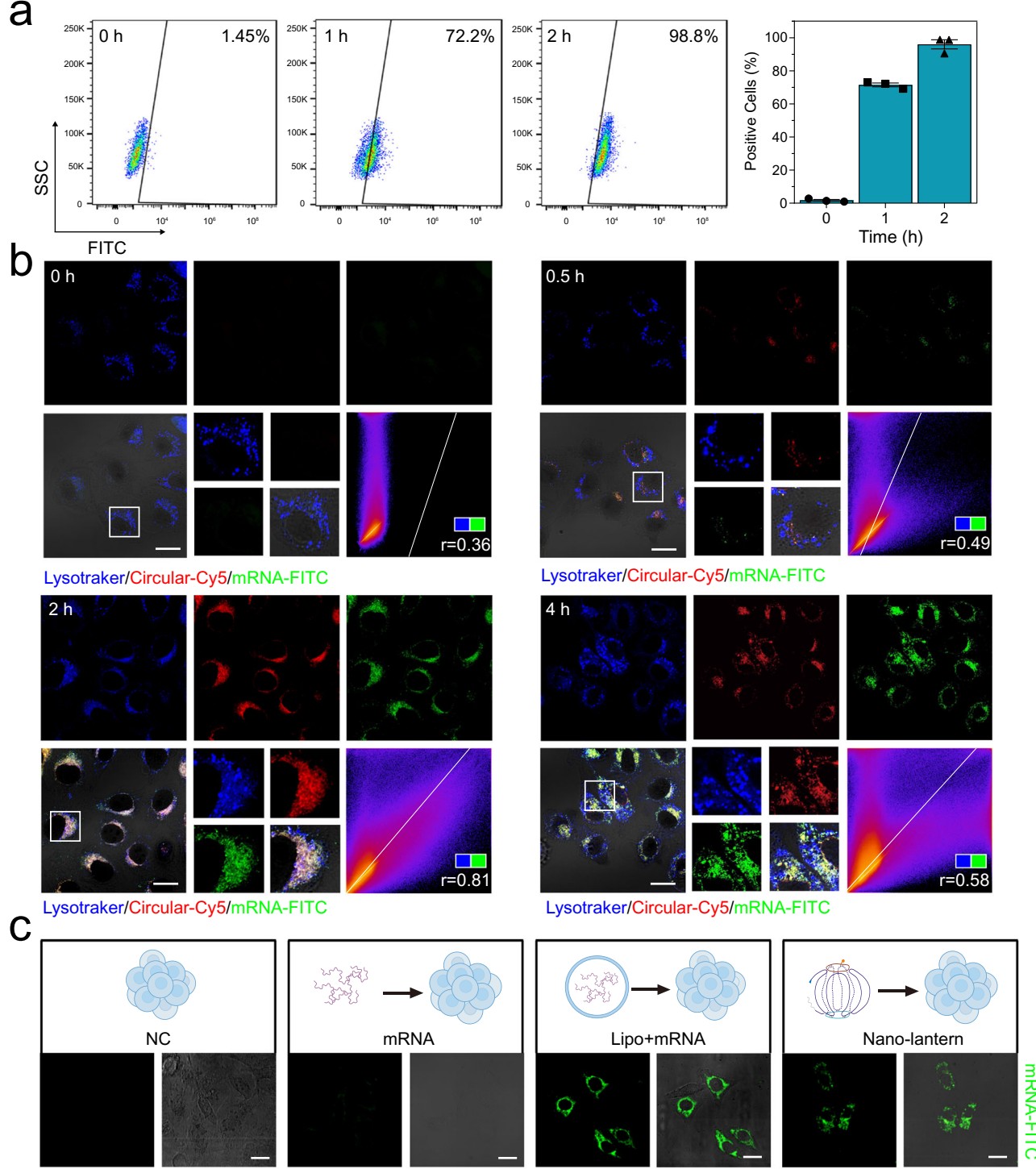

**Fig. 4 | Cellular uptake of *Smad4* mRNA nano-lantern. a** Flow cytometry analysis of the cellular dynamic uptake of mRNA nano-lantern in SW480 cells at different time points, *n* = 3 independent biological samples, data are presented as the mean ± SEM. The gating strategy was provided in the Supplementary Fig. 16b. **b** Representative confocal images of SW480 cells incubated with mRNA nano-lantern at different time points, the images at each time point are from 3 independent experiments. Fluorophore Cy5 was exploited to label the RGD-CS RNA and the FITC was exploited to label the mRNA scaffold. The spearman correlation coefficient (r) is calculated by Image J software to evaluate the association of Lysotraker and mRNA. **c** Representative confocal images of SW480 cells incubated with individual *Smad4* mRNA, lipo + mRNA, and nano-lantern, respectively, from 3 independent experiments. Scale bar: 20 µm. Panel (**c**) created with BioRender.com. Source data are provided as a Source Data file.

macropinocytosis), methyl-beta-cyclodextrin (MBCD, inhibitor of cholesterol-dependent endocytosis), chlorpromazine (CHL, inhibitor of clathrin-mediated endocytosis), and nystatin (NYS, inhibitor of lipid raft-caveolae endocytosis). As shown in Supplementary Fig. 8, after treatment with CHL and NYS, the fluorescence intensity in SW480 cells was significantly reduced, indicating that clathrin or lipid raft-caveolae endocytosis mediated the uptake process of nano-lantern. In addition, a small amount of nano-lantern relied on the cholesterol-dependent endocytosis in the cell due to the slight decline in fluorescence intensity.

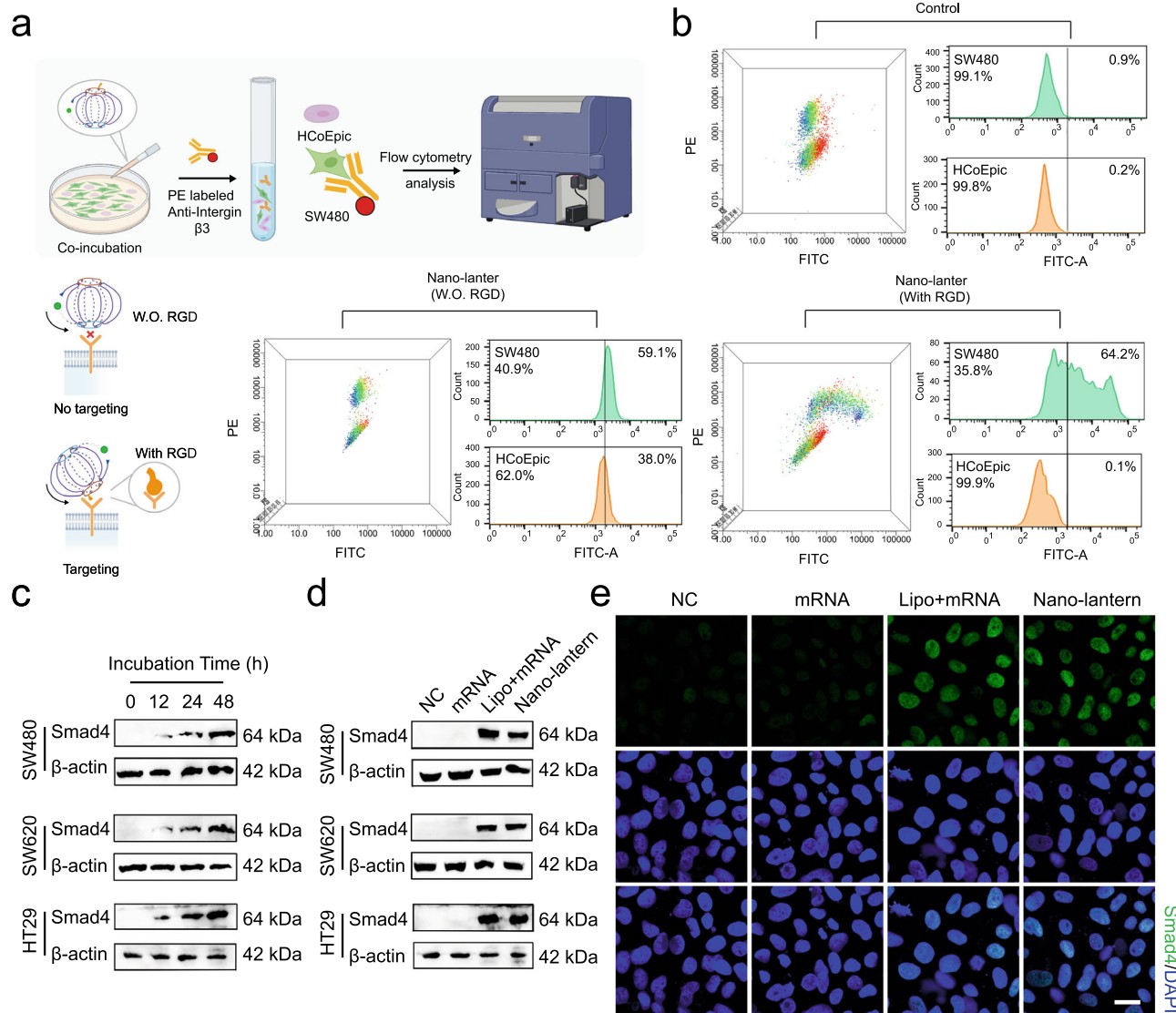

**Fig. 5 | Target ability and intracellular Smad4 expression of nano-lantern in vitro. a** Schematic illustration of targeting experiments by flow cytometry analysis. Created with BioRender.com. **b** Characterization of the targeting ability of *Smad4* mRNA nano-lantern with or without (W.O.) RGD modification by flow cytometry analysis. The gating strategy was provided in the Supplementary Fig. 16c. **c** Representative western blot analysis of Smad4 protein expression in SW480 and SW620 cells treated with mRNA nano-lantern for different incubation times from 3

independent experiments. **d** Representative western blot analysis of Smad4 protein expression in SW480 and SW620 cells treated with individual *Smad4* mRNA, Lipo + mRNA, and nano-lantern from 3 independent experiments. **e** Representative immunofluorescence staining of Smad4 in SW480 cells treated with individual *Smad4* mRNA, Lipo + mRNA, and nano-lantern from 3 independent experiments. Scale bar: 20 µm. Source data are provided as a Source Data file.

Furthermore, we explored the cellular delivery process of nano-lantern. The confocal imaging results were consistent with the flow cytometry results (Fig. 4b), in which fluorophore Cy5 was exploited to label the RGD-CS RNA in addition to the FITC-labeled mRNA scaffold. From the confocal imaging results, it is observed that the mRNA nano-lantern began to be adsorbed on the SW480 cells within 0.5 h; and at 2 h after incubation, most of the nano-lantern was intracellular and colocalized with lysosomes, which were stained with Lysotracker Blue. Cy5-labeled RNA staples and FITC-labeled mRNA were observed to separate from lysosomes after 4 h incubation, suggesting the nano-lanterns escaped from the lysosome and had the availability to liberate mRNA into the cytoplasm. This phenomenon may be attributed to the proton sponge-like mechanism of origami in lysosome. As reported, the origami is rich in $Mg^{2+}$ and can be easily absorbed protons and released metal ions in acidic lysosomes, resulting in the lysosomal escape[33–35]. Western blot results showed the Smad4 expression was markedly decreased in the presence of the proton pump inhibitor

bafilomycin A1 (Baf A1), verifying this hypothesis (Supplementary Fig. 9).

We then compared the efficiency of the nano-lanterns with that of traditional liposomes for intracellular delivery of mRNA. Under the same conditions (RNA concentration, incubation time and buffer solution), the intracellular fluorescence intensity of mRNA of nano-lantern was comparable to the widely used delivery carrier Lipofecta-mine 2000 (Fig. 4c), implying that the lantern-shaped flexible mRNA origami strategy could deliver mRNA to the cytoplasm effectively.

The *Smad4* mRNA nano-lantern is also expected to have targeting capability as the RGD peptide it labeled with can target integrin αVβ3 which is highly expressed in colorectal cancer cells[36,37]. To confirm this speculation, the FITC-labeled nano-lantern with or without RGD modification was incubated with the co-cultured SW480 cells (Integrin αVβ3-positive) and Human Colonic Epithelial Cells (HCoEpiC) (Integrin αVβ3-negative) for 2 h respectively and then probed with PE labeled anti-Integrin β3 (Fig. 5a, Supplementary Fig.10). Flow cytometry was

employed to analyze the targeted delivery of the nano-lantern to objective cells. As shown in Fig. 5b, the 3D flow cytometric plots showed that the nano-lantern without RGD was absorbed by SW480 and HCoEpiC, while the nano-lantern with RGD was uptaken more actively by SW480 (64.2% positive) compared with HCoEpiC (0.1% positive), which was due to the recognition of the RGD peptide and integrin αVβ3. The above results revealed the feasibility of using the RGD modification strategy to realize the targeting ability of the *Smad4* mRNA nano-lantern.

Then, we evaluated the protein translational activity of the nano-lantern in CRC cells. Western blot was exploited to evaluate the Smad4 overexpression level in SW480, SW620 and HT29 cells. As shown in Fig. 5c, the expression level of Smad4 protein increased significantly within 24 h after transfection and reached the maximum at 48 h. In addition, we also investigated the overexpression efficiency of Smad4 in CRC cells treated with different transfection reagents. As shown in Fig. 5d and Supplementary Fig.11, compared with lipofectamine, when the same amount of mRNA was delivered with nano-lanterns, the protein expression intensity reached 89% in SW480 cells, 94% in SW620 cells. Similar overexpression of Smad4 in HT29 cells was observed, demonstrating a non- cell-type specific effect with another independent CRC line. Immunofluorescence images showed similar results that Smad4 was significantly overexpressed in SW480 and SW620 cells treated with nano-lanterns and Lipofectamine 2000 transfection (Lipo+mRNA) compared with individual mRNA incubation. Moreover, we found that Smad4 protein as a nuclear protein was mainly located in the nucleus after exogenous transfection (nano-lantern or Lipo+mRNA) (Fig. 5e). Western blot results demonstrated the nuclear translocation of Smad4 after nano-lantern transfection. As shown in Supplementary Fig.12, the protein expression level of Smad4 increased obviously in the cytoplasm and nucleus after transfected with the *Smad4* nano-lantern. Quantitation of the nuclear and cytoplasmic western blot signals indicated that the nuclear-to-cytoplasmic ratio of Smad4 at 24 h post-transfection with *Smad4* nano-lantern increased. Collectively, these results indicated that the ectopic expression of Smad4 could endow the homing ability, which serves as a prerequisite for its intrinsic biological function.

**The tumor suppression of Smad4 mRNA nano-lantern in vitro**
Before the suppression evaluation of *Smad4* mRNA nano-lantern on CRC cell lines, the cytotoxicity of these nano-carriers should be investigated. The prone to degraded uncapped and untailed mRNA (Uu-mRNA) was adopted to avoid up-regulating of Smad4 protein and suppressing the interference of the expression of Smad4 protein to the cell viability (Supplementary Fig. 13). Then, we used liposomes and nano-lanterns to carry (Uu-mRNA) in proportion for cell treatment to investigate the cytotoxicity of nano-lantern and liposomes. As shown in Fig. 6a, compared with nano-lantern, Lipofectamine 2000 showed significant cell toxicity at the mRNA concentration of 0.125 μg/μl, and with the increase of Lipofectamine 2000 and mRNA concentration, Lipofectamine 2000 exhibited higher toxic; in contrast, no obvious cytotoxicity was observed in the nano-lantern group.

The tumor suppressive properties of *Smad4* mRNA nano-lantern on CRC cell lines including effects on proliferation, clonogenicity and migration were then investigated. SW480 and SW620 cells were treated with different formulations (individual mRNA and nano-lantern). The cell growth of treated SW480 and SW620 was first evaluated. Results of the cell counting kit-8 (CCK-8) assay indicated that compared with individual mRNA, the proliferation ability of cells treated with nano-lanterns was significantly inhibited (Fig. 6b). Similarly, the EdU assay suggested that nano-lantern treatment could reduce cell growth in SW480 and SW620 cells (Fig. 6c).

Besides, in vitro colony formation assay displayed that overexpression of Smad4 after nano-lantern treatment inhibited the clonogenicity of CRC cells (Fig. 6d). The results of scratch wound assay

and transwell invasion assay in vitro further showed that ectopic expression of Smad4 suppressed the mobility and invasiveness of CRC cells (Fig. 6e, f). Altogether, we noted a decrease in proliferation, clonogenecity and migration as supportive of the known tumor suppressor effects of Smad4 in CRC[3–7].

**Smad4 directly binds to the MYC promoter and regulates MYC expression in CRC cells**
The Smad4 family has been reported to function as a CRC suppressor potentially participating in transcription inhibition of several target oncogenes, such as c-MYC, VEGFC, CXCL5, and so on[5–7]. The c-MYC proto-oncogene (MYC) identified mutations and expression changes in CRC, which has served as a clinical therapeutic target[38,39]. Here, we further investigated whether the upregulated Smad4 protein induced by nano-lanterns could play the transcription inhibition on MYC to suppress the proliferation, mobility, and invasion of CRC cell lines. According to the JASPAR databases, we found the Smad4 potential binding site (Fig. 7a). By analyzing the MYC promoter, we found that Smad4 members could bind to the −805 ~ −817 bp of the MYC promoter (Fig. 7b). To illustrate the enrichment of Smad4 on the MYC promoter region, the Smad4 in cells were overexpressed by nano-lantern (Fig. 7c), and then the ChIP-qPCR assay was exploited to indicate that Smad4 was recruited to binding to the MYC promoter at −805 ~ −817 bp (Fig. 7d and e). The above results indicated that Smad4 was recruited to the binding site of the MYC promoter and involved in the transcription regulation of MYC.

To further clarify whether Smad4 down-regulates MYC, we probed the protein profile changes of MYC by western blot after Smad4 overexpression induced by nano-lantern. The western blot results showed that in Smad4-overexpressing CRC cells induced by nano-lantern, MYC was downregulated (Fig. 7f). In the meanwhile, Smad4 could not be significantly overexpressed in the cells treated with mRNA alone, so there was no significant downregulation on downstream of MYC. To demonstrate the regulation of Smad4 on other oncogenes, we also examined expression changes of oncogene-related molecules, VEGFC and CXCL5 by western blot after Smad4 overexpression. The results showed that VEGFC and CXCL5 were also negatively regulated by Smad4 (Fig. 7g). In addition, the downstream genes also changed with the nuclear translocation of Smad4 (Supplementary Fig. 12). Furthermore, after removing the nano-lantern, the protein expression of Smad4 declined and downstream genes regulated by Smad4 was recovered, which manifested the regulation function of Smad4 toward these genes (Supplementary Fig. 14).

**The tumor suppression of mRNA nano-lantern in a subcutaneous model**
The tumor suppression of mRNA nano-lantern in vivo was then investigated through a BALB/c nude mice xenograft model, established by planting SW480 cells. After 21 days of implantation, nude mice were randomly divided into three groups. Each group was subcutaneously injected around the tumor tissues with saline, individual mRNA, and mRNA nano-lantern every 3 days for eight doses, respectively (Fig. 8a). As shown in Fig. 8b–e, the mRNA nano-lantern efficiently suppressed the tumor growth compared with the saline-treat and the individual mRNA-treat groups with a smaller volume of the tumor. Next, we collected the tumor tissues and internal organs of nude mice for immunohistochemistry (IHC) and histopathological (H&E) staining. IHC staining demonstrated that the *Smad4* mRNA nano-lantern successfully restored the Smad4 protein expression in tumor tissues, which intuitively suggested the successful translation of nano-lantern in tumor tissues while the bare mRNA could not. Histopathological observation also showed that no apparent damages were observed in the parenchyma organs of tumor-bearing mice in all groups (Fig. 8g), which exhibited the low toxicity of the *Smad4* mRNA nano-lantern. To further analyze the mechanisms of the tumor

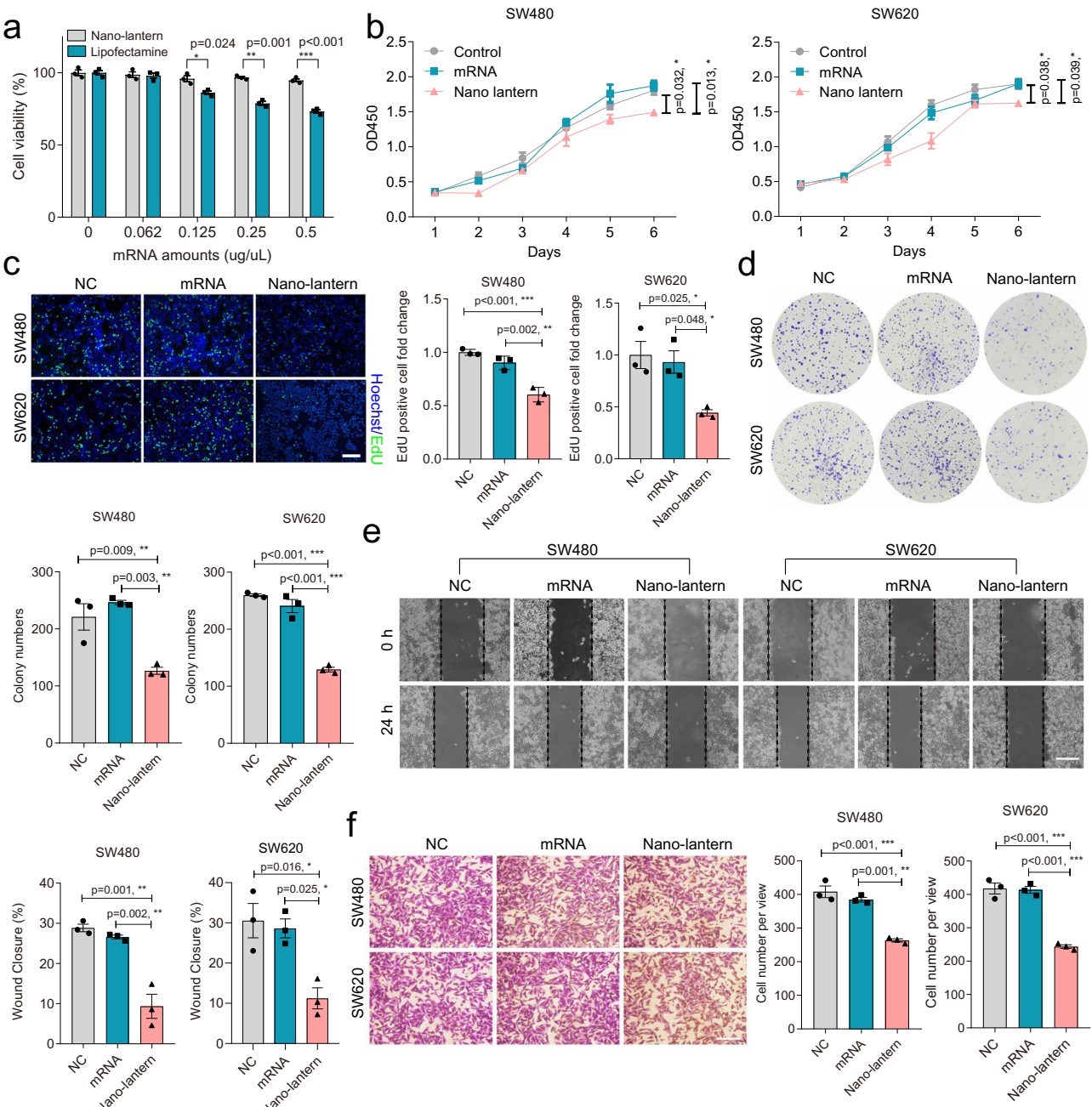

**Fig. 6 | *Smad4* mRNA nano-lantern introduction correlates with reduced proliferation, clonogenicity and migration in vitro. a** Cell viability assays of SW480 cells treated with Lipo+Uu-mRNA and nano-lantern with Uu-mRNA for 24 h, respectively. **b–d** The cell proliferation ability of SW480 and SW620 cells incubated with individual *Smad4* mRNA or mRNA nano-lantern by (**b**) CCK8 assay, (**c**) EdU assay, Scale bar: 100 µm, and (**d**) plate colony formation assay. **e** The cell migration ability of SW480 and SW620 cells incubated with individual *Smad4* mRNA or mRNA nano-lantern by a scratch wound assay. Scale bar: 100 µm. **f** The cell invasion of SW480 and SW620 cells was incubated with individual *Smad4* mRNA or mRNA nano-lantern by in vitro transwell invasion assay. Scale bar: 50 µm. Data are presented as the mean ± SEM. *n* = 3 independent biological samples. Panel (**a**), independent-sample *t*-test, two-sided, statistical differences of the panel (**b**–**f**) were assessed using one-way ANOVA with Bonferroni multiple comparisons test. **p* < 0.05, ***p* < 0.01, ****p* < 0.001. Source data are provided as a Source Data file.

inhibition ability of mRNA nano-lantern, we evaluated the expression level of Smad4, downstream transcriptional regulatory genes, MYC, VEGFC, and CXCL5 in tumor tissues by Western blot. Similar to the results at the cellular level (Fig. 7f, g), here we found that mRNA nano-lantern promoted the expression of Smad4 protein in tumor tissues and induced the decline of MYC, VEGFC, and CXCL5 (Fig. 8h). Together, the above results indicated the successful suppression of tumor growth and low toxicity to parenchyma organs after treatment with mRNA nano-lantern in vivo, which is also consistent with the inhibition effect in vitro.

**The tumor suppression of mRNA nano-lantern in an orthotropic cecal injection model**

An orthotopic colorectal tumor model was further established to simulate the tumor microenvironment, and the tumor suppression of nano-lantern in an orthotopic colorectal tumor model was then investigated (Fig. 9a). We used the luciferase-lentiviral transduction to facilitate bioluminescence monitoring of tumor size incidence. To visualize the biodistribution of the nano-lantern in the orthotopic model, the Cy5-labeled nano-lantern in mice was imaged using an IVIS Spectrum of the small-animal imaging system (Fig. 9b). The in vivo

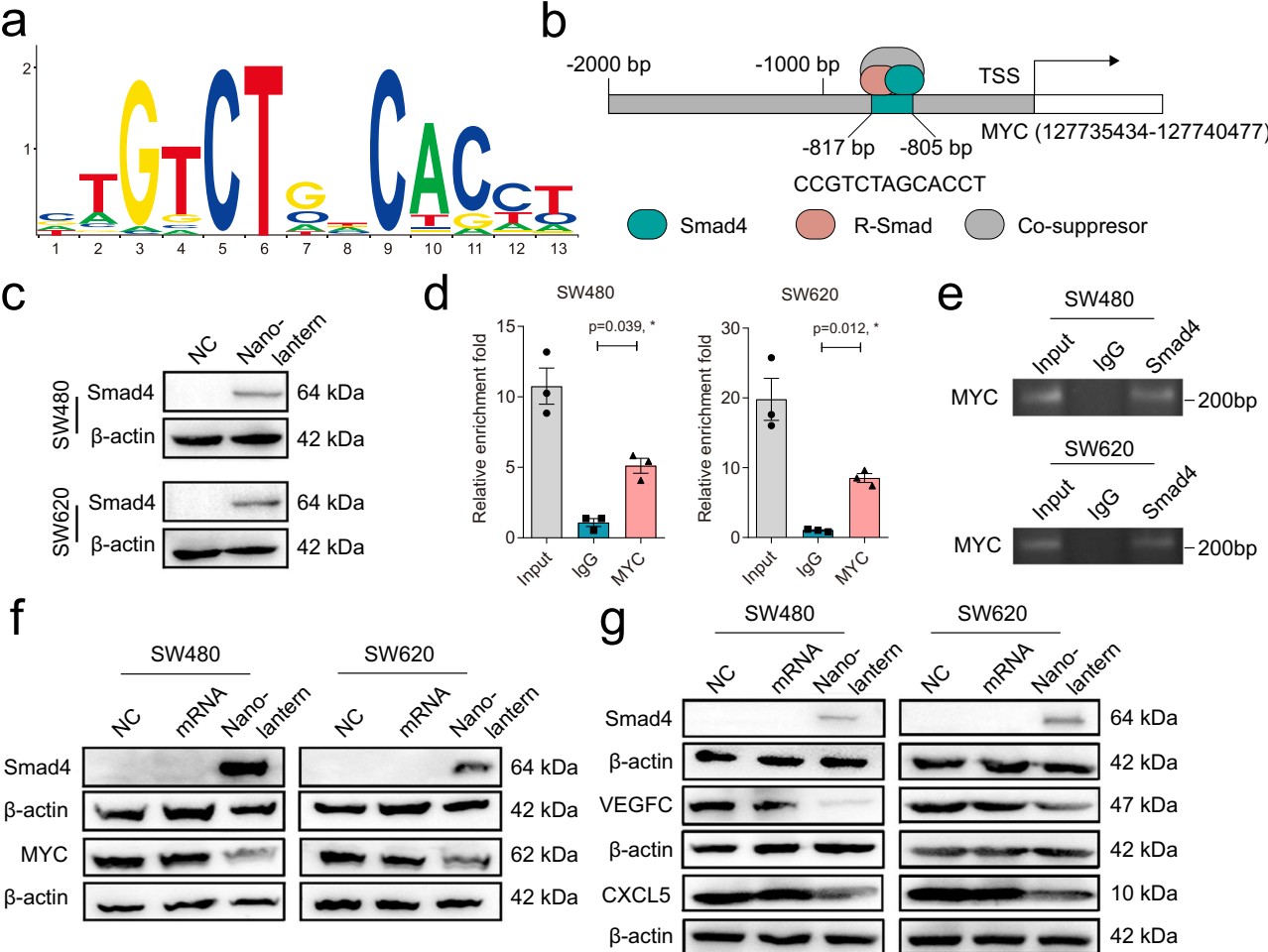

**Fig. 7 | Smad4 transcriptional suppressed the expression of MYC in CRC cells.**
**a** The Smad4 potential binding site in JASPAR software. **b** Potential recognition sites of Smad4 in the MYC promoter region. **c** Western blot analysis of Smad4 expression in SW480 and SW620 cells before and after treatment with nano-lantern. **d** ChIP assays with anti-Smad4 or negative control (anti-IgG) antibodies showed Smad4 binding to the recognition site ($n = 3$ independent biological samples).
**e** Representative agarose electrophoresis for ChIP analysis of Smad4 binding to the MYC promoter from 3 independent experiments. **f** Representative western blot

analysis of MYC expression in SW480 and SW620 cells after treatment with mRNA, and nano-lantern from 3 independent experiments. **g** Representative western blot analysis of VEGFC and CXCL5 expression in SW480 and SW620 cells after treatment with mRNA, and nano-lantern from 3 independent experiments. Data are presented as the mean ± SEM. Statistical differences were assessed using one-way ANOVA with Bonferroni multiple comparisons test. *$p < 0.05$, **$p < 0.01$, ***$p < 0.001$. Source data are provided as a Source Data file.

images showed the strong Cy5 fluorescence signals in the abdomen at first and then faded with times after intraperitoneal injection the formulations. For the nano-lantern group, after 4 h injection, the fluorescence signals were still distinct which overlapped with luciferase luminescence signals of SW480-Luc tumor cells, while the fluorescence of mRNA become invisible. These results exhibited a better tumor retention ability of nano-lantern in vivo compared to bare mRNA.

Ex vivo images of tumors displayed the smaller orthotopic xenografts in the nano-lantern group compared with bare mRNA groups, which suggested the decreased tumorigenicity (Fig. 9c). In addition, there were no significant changes in the weight of mice and the proinflammatory factors IFN-β and IL-6, demonstrating the biocompatible and low immunogenic property of nano-lantern (Fig. 9d and Supplementary Fig. 15). Next, the tumor xenograft tissues and tumor-bearing mouse organs were collected for IHC and H&E staining. The IHC analysis showed successful overexpression of Smad4 in tumor lesions after being treated with nano-lantern, and H&E images showed normal morphology in the mouse parenchyma organs and no metastases in liver or lung were noted (Fig. 9e, f). The protein expression profiles in xenografts displayed the negative correlation of Smad4 and

its downstream genes, MYC, VEGFC, and CXCL5 (Fig. 9g). The above results demonstrated the mRNA nano-lantern could realize overexpression of Smad4 in orthotopic tumor models and inhibit their growth successfully.

## Discussion

The loss and mutation of tumor-suppressor genes are the main reason for tumor progression and clinical resistance to various therapeutic methods[40,41]. Reversal of the phenotype induced by loss or mutation of tumor suppressors has been proven to be an elusive goal[42,43]. For the treatment of solid tumors, mRNA-based therapeutics, though also showing great promise, have encountered some resistance due to their suboptimal efficacy. The difficulty mainly comes from two aspects. One is the selection of the cargo mRNA. Literature have shown that *Smad4* is a tumor suppressor gene in CRC, which may serve as a therapeutic candidate[3–7]. Recently, Xinwei Liu also demonstrated that Smad4 can reinforce T cell function by promoting CD8 + T cell-mediated cytotoxic immunity[44]. Thus, we actively explored this aspect by selecting *Smad4* mRNA as a cargo that had never been tried before. We showed that after the delivery of *Smad4* mRNA, the downstream oncogenes can be transcriptional regulated effectively, and significant

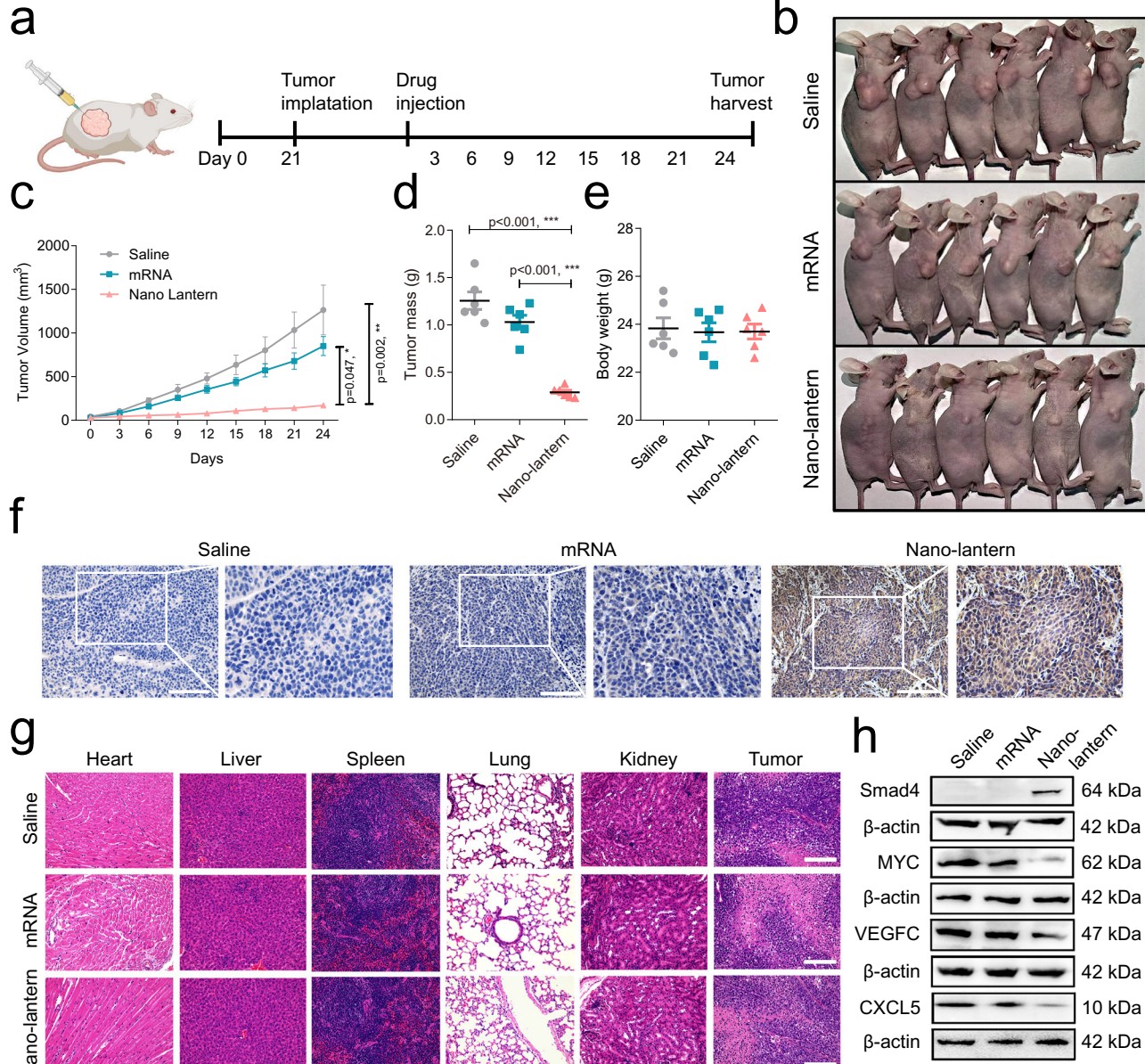

**Fig. 8 | Tumor suppression of *Smad4* mRNA nano-lantern in the subcutaneous model. a** Systematic schedule of Smade4 mRNA nano-lantern administration in the subcutaneous model of SW480 cells. Created with BioRender.com. **b** Photographs of the subcutaneous tumor-bearing nude mice after treatment with saline, individual mRNA, and mRNA nano-lantern, respectively. **c** Images of the xenograft tumors at the endpoint (*n* = 6 mice/group). **d** Tumor growth curves of the tumor-bearing mice after treatment with saline, individual mRNA, and mRNA nano-lantern, respectively (*n* = 6 mice/group). **e** Body weight and tumor weight changes in the tumor-bearing mice with different treatments (*n* = 6 mice/group).

**f** Representative immunohistochemical staining for Smad4 in tumor tissues from 6 mice/group. Scale bar: 100 μm. **g** Representative H&E staining assay of organs and tumor tissues in the tumor-bearing mice of each group from 6 mice/group. Scale bar: 100 μm. **h** Representative western blot analysis of Smad4, MYC, VEGFC, and CXCL5 expression in tumor tissues from 6 independent experiments. Data are presented as the mean ± SEM. Statistical differences were assessed using one-way ANOVA with Bonferroni multiple comparisons test. *$p < 0.05$, **$p < 0.01$, ***$p < 0.001$. Source data are provided as a Source Data file.

suppression of colorectal cancer growth was observed both in vitro and in vivo.

Regarding the low stability of mRNA in vitro, a carrier is exploited for mRNA intracellular delivery. While the use of carriers may bring new problems, such as immunogenicity and toxicity. Thus, the vector-free delivery strategy in the development of mRNA drugs is getting attention. For example, Naoto Yoshinaga proposed a concept of mRNA delivery by bundling mRNA strands with RNA oligonucleotide linkers[29]. This mRNA nano structure boosted its RNase resistance ability and expressed the target protein in vivo without using additional vectors.

In this research, we explored the probability of mRNA delivery based on RNA origami technology. As a highly programmable self-assembling nanotechnology, DNA or RNA origami make it possible to construct scalable nucleic acid nanostructure which holds the characteristics of high stability for a large range of applications including drug delivery. However, due to the fully double-strand and rigid nanostructure of traditional origami that cannot be dehybridized back to a free single-stranded state, it seems not suitable for mRNA delivery. Here, we propose a concept of flexible origami that not only incorporates the rigid structure of traditional origami, but also retains

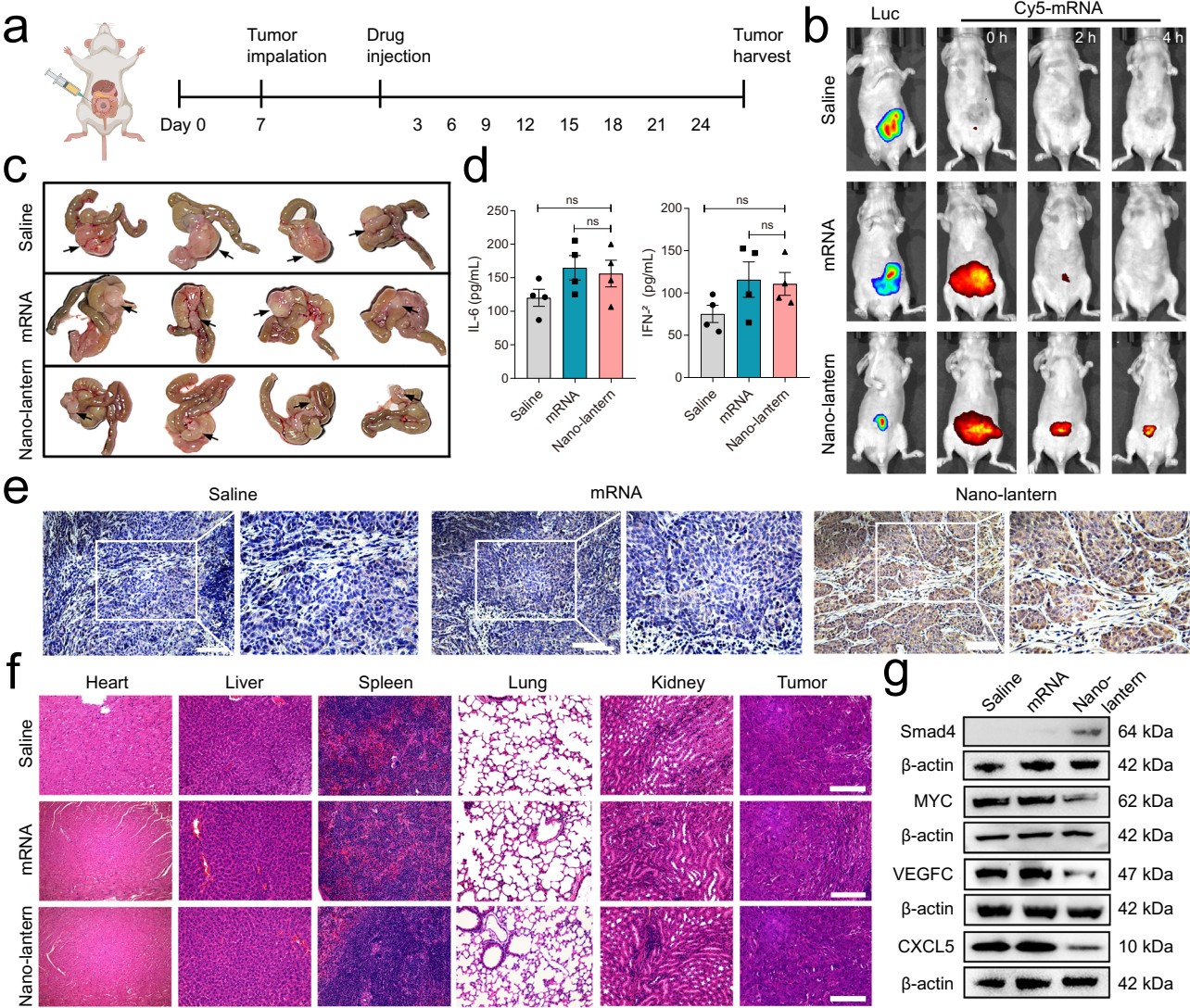

**Fig. 9 | The tumor suppression of mRNA nano-lantern in an orthotropic cecal injection model. a** Systematic schedule of Smade4 mRNA nano-lantern administration in the orthotopic model of SW480-Luc cells. Created with BioRender.com. **b** Representative bioluminescence imaging of tumor burden and fluorescence imaging of Cy5-mRNA nano-lantern in tumor-bearing nude mice. **c** Ex vivo images of the xenograft tumors at the endpoint. **d** IL-6 and INF-β levels in peripheral blood of tumor bearing mice after 6 h post-intraperitoneal injection of different formulations (*n* = 4 mice/group). **e** Representative IHC staining for Smad4 in tumor tissues from 4 mice/group. Scale bar: 100 μm. **f** Representative H&E staining of organs and tumor tissues in the tumor-bearing mice from 4 mice/group. Scale bar: 100 μm. **g** Representative western blot analysis of Smad4, MYC, VEGFC, and CXCL5 expression in orthotopic tumors from 4 independent experiments. Data are presented as the mean ± SEM. Statistical differences were assessed using one-way ANOVA with Bonferroni multiple comparisons test. *$p < 0.05$, **$p < 0.01$, ***$p < 0.001$. Source data are provided as a Source Data file.

a large number of flexible unstapled parts. The former enables nanolization, while the latter allows the functionality of single-strands. On the premise of ensuring nanolization, as few staples as possible may be the key. Here, according to the characteristics of mRNA, we only use two circular RNAs as staples to realize the mRNA origami, namely lantern-shaped flexible origami, which may be the least used staple at present. Our results show that staple-assisted mRNA origami nanolization promotes cellular endocytosis, while the relatively flexible structure allows the mRNA to still be recognized and translated. We also demonstrated that without using exogenous chemicals, the mRNA nano-lantern exhibited better biocompatibility, lower toxicity, and a similar level of mRNA delivery ability. Compared with the existing delivery carriers, our scheme realizes the facile nanolization of single mRNA molecule, which may help to modify, control and expand the structure and function of mRNA molecules, e.g. directional modification of mRNA molecules with functional domains, quantitative delivery of mRNA with single molecular precision, etc.

Although the initial exploration of the mRNA nano-lantern exhibited acceptable performance, there is still much room for improvement in this scheme. The improvement of mRNA stability by this strategy is still limited. In addition, the binding sites are also strictly confined, which is closely related to transfection efficiency and translation activity. Future concerns could focus on exploring the use of diverse flexible origami structures to deliver different mRNAs and even developing computer-aided design and artificial intelligence simulations in the structure and function of the flexible origami. Thereafter, this strategy may be developed as an alternative delivery strategy for mRNA-based therapy.

## Methods
### Ethical statement
This research complies with all relevant ethical regulations. All the experiments in this research were approved by the Ethics Committee of Shanghai University (No.2022-238).

## Materials

The plasmid containing the *Smad4* gene and the T7 promoter was provided by Hunan Fenghui Biotechnology Co., Ltd. (Changsha, China). All RNA oligonucleotides were synthesized and purified by Shanghai Generay Biotech Co., Ltd (Shanghai, China). The sequences are shown in Supplementary Table 1. The antibodies are displayed in Supplementary Table 2. PCR purification kit, MEGAclear kit, DNase I, EcoRI enzyme, and all RNase-free pipet tips and centrifugal tubes were purchased from Thermo Fisher Scientific Co., Ltd (MA, USA). Hifair® T7 High Yield RNA Synthesis Kit and Vaccinia Capping Enzyme were obtained from Yeasen Biotechnology Co., Ltd (Shanghai, China). All other chemicals with analytical grades were provided by Nanjing KeyGen Biotech Co., Ltd. (Nanjing, China).

## In vitro transcription (IVT) mRNA

IVT method was used to synthesize mRNA. Briefly, the templated plasmid containing the T7 promoter and the open-reading frame of the *Smad4* gene was cleaved into linearity using the EcoRI enzyme, followed by purification of the digestion products with a PCR purification kit. The mRNA was synthesized using the Hifair T7 High Yield RNA Synthesis Kit according to its manual and then purified the mRNA using a MEGAclear kit. The naked mRNA was capped with an m7G cap in its 5′ end using the Vaccinia capping enzyme and then added a poly (A) tail at its 3′ end by using the polyadenylic acid polymerase. The capped and tailed mRNA was then purified by the MEGAclear kit for further use.

## Circular staple RNA (CS RNA) Synthesis

The linear staple RNA was mixed with splint RNA at a ratio of 1:1.5 and then added $1 \times$ T4 RNA ligase buffer. The solution was incubated at 95 °C for 5 min, then slowly cooled down to room temperature. After low-speed centrifugation, T4 RNA ligase was added to the mixture to incubate at 37 °C for 4 h, followed by heating at 65 °C for 15 min to inactivate the ligase.

## RGD labeled CS RNA synthesis

RGD and circular staple RNA (CS RNA) was conjugated through Sulfo-SMCC reaction. In detail, 92 μL of 1 mM thiol-CS RNA was mixed with 4 μL of 30 mM TCEP and 4 μL of 1 M sodium phosphate buffer at room temperature for 1 h. The mixture was then purified with an Amicon-10K. 1 mg of Sulfo-SMCC was added to 50 μL of 10 mM RGD and vortexing at low speed for 5 min and then kept at room temperature for 1 h. The RGD solution was then centrifuged to remove the insoluble Sulfo-SMCC. Subsequently, the Sulfo-SMCC activated RGD solution was added to the TCEP-treated thiol-modified RNA. After gently mixing, the mixture was kept at 4 °C overnight and then purified through an Amicon-10K to remove the remaining RGD.

## Lantern-shaped flexible origami preparation

*Smad4* mRNA and CS RNA were mixed at a molar ratio of 1:10:10 in $1 \times$ TAE/Mg$^{2+}$ buffer (40 mM Tris, 20 mM acetic acid, 2 mM EDTA and 12.5 mM magnesium acetate). After gently mixing, the mixture was heated at 65 °C for 10 min and then cooled down to room temperature in 2 h. The lantern-shaped flexible origami was characterized by agarose gel electrophoresis.

## Cell-free translation

Human Cell-Free Protein Expression System (Takara, Dalian, China) was used to analyze the translation ability of mRNA nano-lantern according to its protocol. Briefly, naked *Smad4* mRNA, mRNA nano-lantern (3 bs, 5 bs, 7 bs) with the same concentration of 25 ng/μL was incubated in the solution containing HeLa Lysate, Accessory Proteins, Reaction Mix for 6 h at 30 °C, respectively. Western blot analysis was adopted to detect the expression of the targeted protein.

## Cell culture

The human CRC cell lines SW480 (CCL-228), SW620 (CCL-227), HT29 (HTB-38) were purchased from the American type culture collection (ATCC), human colonic epithelial cell (HCoEpiC, 2950) was obtained from ScienCell Research Laboratories, and cultured in DMEM containing 10% fetal bovine serum, 1% penicillin and 1% streptomycin at 37 °C in an atmosphere with 5% CO$_2$.

## AFM imaging

A 10 μL sample containing 10 mM MgCl$_2$ was deposited onto cleaved mica for 30 s, washed with DEPC H$_2$O for about 15 s gently, then dried with compressed air at room temperature. MultiMode 8 AFM (Bruker) was used to characterize the samples in air phase mode.

## Dynamic light scattering

ZETASIZER 3000HS instrument (Malvern Instruments Ltd., UK) was used to characterize the size distributions of the nucleic acid nanostructure.

## Confocal imaging analysis

The cells were seeded in the confocal dishes overnight and then treated with the lysotracker to label the lysosome. After then, the cells were incubated with FITC/Cy5-labeled nano-lantern for 0, 0.5, 2, 4 h, respectively. Subsequently, removed the supernatant and washed the cell with $1 \times$ PBS three times. After then, a Zeiss scanning microscope (Carl Zeiss LSM710, Germany) was used to image the fluorescence signals. The co-location of lysotraker blue and mRNA-FITC was statistically analyzed by image J software.

## Flow cytometry analysis

Cells were cultured in the 12 well plates and treated with FITC labeled mRNA nano-lantern for different time points (0, 1, 2 h) followed by washing with PBS three times. The cells were collected for cytometry analysis.

For cell targeting experiments, the HCoEpiC and SW480 were co-incultured in a 12 well plates and treated with FITC labeled mRNA nano-lantern for 2 h. Then, the cells were harvested and probed with PE-Intergrin β3 antibody, which was employed to lable the Intergrin β3 positive CRC cells. The cells were washed with PBS three times and then collected for flow cytometry analysis. The percentage of positive cells was evaluated by Flowjo software.

## Immunofluorescence staining

The cells were planted in confocal dishes and cultured overnight, followed by fixing with 4% paraformaldehyde at room temperature. Cells were then permeabilized with 0.2% Triton X-100-PBS for 10 min and further incubated with PBS blocking buffer (2% goat serum, 2% BSA, 0.2% gelatin) for 1 h. Subsequently, the cells were incubated with primary antibody for 2 h (1:500) at room temperature, then washed with PBS and probed with FITC labeled secondary antibody in blocking buffer (1:500) for 1 h at room temperature. The samples were washed in PBS and then treated with the DAPI in PBS (1:2000) to stain nuclei. After that, the cells were imaged under the confocal microscope.

## Western blot analysis

Total proteins from SW480/SW620 cells and tumor tissues were extracted using RIPA buffer (KeyGEN, China) with 1% PMSF (KeyGEN). The BCA Protein Assay Kit (TaKaRa) was used to determine the protein concentration. Protein samples were separated on 10% SDS-PAGE gel, transferred onto polyvinylidene difluoride membranes, and incubated with the primary antibodies respectively at 4 °C overnight. Then the membranes were blotted with secondary antibodies at room temperature for 2 h. Enhanced chemiluminescence

was used to visualize the protein bands in the ChemiDoc XRS System. And the β-actin served as an overall internal reference and histone H3 served as a nuclear internal reference. The dilution of antibodies was displayed in Supplementary Table 2.

## Colony formation assay
SW480/SW620 cells treated with naked *Smad4* mRNA or nano-lantern were cultured in 6 well plates for 2 weeks. 4% phosphate-buffered formalin was used to fix the colonies, and then the colonies were stained with Giemsa solution for 20 min.

## Wound healing assay
Pretreated cells were planted in 6 well plates for about 24 h. The middle of the cells in each plate was scratched using a sterilized 200 μL pipette tip, followed by washing with 1 × PBS and then adding 2 ml DMEM containing 2% FBS and 1 mM thymidine (Sigma-Aldrich, USA) into each well. Monitoring the scratches width at 0 and 24 h, respectively.

## Migration assay
Transfected cells were cultured in the upper chamber of Matrigel (BD Biosciences, CA, USA) coated transwell filter champers with serum-free DMEM. The bottom chamber was added DEMD with 10% FBS. After 24 h of culture, we used the sterilized cotton swab to remove the non-migrated cell in the upper chamber. The cells on the bottom chamber were fixed with 4% phosphate-buffered formalin, followed by staining with Giemsa for 25 min, and then count the cells in each well.

## EdU incorporation assay
SW480/SW620 cells were seeded in 96 well plates at a density of $2 \times 10^5$ cell /mL and transfected with naked *Smad4* mRNA and nano-lantern, respectively. After 48 h incubation, EdU was added into each well for co-culture 2 h. The fluorescent microscope was exploited to record the EdU and Hoechst positive cells.

## Chromatin immunoprecipitation assay
The Ch-IP assay (Millipore, MA, USA) was adopted to study the DNA-protein interactions according to its manufacturer's protocol. Briefly, SW480 and SW620 cells were cross-linked through formaldehyde, followed by quenched through glycine. Then, the cells were washed with 1 × PBS twice and centrifuged at $800 g$ for 5 min. Subsequently, the cells were resuspended in lysis buffer for 15 min at 4 °C, followed by centrifuging at $800 g$ for 5 min, and then resuspended in nuclear lysis buffer. After sonication, the supernatant was added with antibodies (IgG/Smad4) and magnetic beads and rotated slowly overnight at 4 °C. After the magnetic beads were cleaned, eluted, and purified, the DNA samples were collected for further qPCR and electrophoresis analysis.

## Subcutaneous and orthotopic tumor xenograft models
All experiments of animal were in accordance with the guidelines on the use and care of laboratory animals for biomedical research published by the National Institutes of Health, and approved by the Ethics Committee of Shanghai University (No. 2022-238). Female BALB/c nude mice (purchased from Shanghai SLAC Laboratory Animal Company, Shanghai, China), 4–5 weeks, housed in a barrier facility on a 12 h light/dark cycle at 22–24 °C and 45–55% humidity. The maximal tumor size is limited to 2 cm in length and the maximal tumor burden was not exceeded in all the animal experiments. For the subcutaneous tumor model, each nude mouse was subcutaneously inoculated with $2 \times 10^6$ SW480 cells under aseptic conditions. After 3 weeks of inoculation, the nude mice were divided into three groups randomly and injected with saline, naked *Smad4* mRNA, and nano-lantern, respectively. For the orthotopic

xenograft tumor model, BALB/c nude female mice were anesthetized through isoflurane. $2 \times 10^6$ SW480-Luc cells were mixed with matrigel (1:1, $v/v$) and then were inoculated into the cecal wall with an insulin gauge syringe through a midline incision. IVIS system was adopted to monitor the tumor growth via tail vein injection of D-luciferin (10 mg/mL). After the luminescence intensity reach about $1.0 \times 10^9$, interventions started followed by intraperitoneal injection of saline, naked *Smad4* mRNA, and nano-lantern, respectively. Mouse body weight and tumor volumes (smaller diameter$^2$ × larger diameter × 0.5) were measured twice a week. After the tumor was harvested, tumor tissues and organs, including the heart, liver, spleen, lung, and kidney, were fixed in formaldehyde solution for H&E staining. Proteins were extracted from the tumor tissues for Western blot analysis.

## ELISA
Serum samples from mice were collected in 1.5 mL EP tubes and analyzed for mouse IFN-β, IL6 by enzyme-linked immunosorbent assay (Solarbio, SEKM- 0032, SEKM-0007) according to its manufacturer's instruction.

## Statistics and reproducibility
All data are presented as mean ± SEM using GraphPad Prism 8.0. for bar graphs. Statistical significance was analyzed using SPSS 26.0 software (SPSS Inc., Chicago, IL). The comparison between the two groups was conducted by the independent-sample $t$-test. One-way ANOVA with Bonferroni test was used to compare multiple groups. A $p$-value of <0.05 was considered statistically significant for all datasets. The sample sizes are provided in the figure legends. No statistical method was used to predetermine the sample size. Samples were randomly allocated in the study. The investigators were blinded to sample collection and data analysis. All experiments were conducted in at least triplicates independently to guarantee reproducibility.

## Reporting summary
Further information on research design is available in the Nature Portfolio Reporting Summary linked to this article.

# Data availability
The potential recognition sites of Smad4 in the MYC promoter region was obtained from JASPAR database (http://jaspar2016.genereg.net). All data generated or analyzed during this work are included in this article and its Supplementary Information files. Source data are provided with this paper.

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

## Acknowledgements

This work was supported by the National Natural Science Foundation of China (Grant Nos. 32150011 and 22074090 to Xiaoli Zhu, 81930066 to Fenyong Sun), the Natural Science Foundation of Shanghai (19ZR1474200 to Xiaoli Zhu) and the Clinical research project of Shanghai Tenth People's Hospital (YNCR2A005 to Xiaoli Zhu).

## Author contributions

F.S., and X.Z. conceived the research studies. B.G. contributed materials and analysis tools, M.H. and C.F. conducted experiments, Q.Y. and C.L. performed data analyses. M.H. and C.F. wrote the manuscript. X.Z. supervised the work. All authors contributed to the editing of the manuscript.

## Competing interests

The authors declare no competing interests.
