## [Peer Review File · Nature Communications]

REVIEWER COMMENTS

Reviewer #1 (Remarks to the Author):

This manuscript addresses the use of RNA nano-structure for mRNA delivery without using mRNA delivery nanoparticles. mRNA was hybridized with circular RNA at 3 – 7 sites to prepare a lantern-shaped structure, and a targeting ligand was attached to circular RNA. The structuring improved mRNA nuclease stability, with the ligand enabling successful targeting. Without the use of delivery nanoparticles, their system is non-cytotoxic. The system was then used for delivering Smad4, a tumor suppressor gene, showing successful outcomes in vitro and in vivo. The concept is unique, and the experiments are well-designed. This manuscript is acceptable after addressing the following points.

- 1) Why was the hybridization length determined to be 17 nt?
- 2) It is interesting to find that 3 bs and 5 bs showed a large difference in compaction status. However, the difference in serum stability (Fig. 2a) is not large. Factors other than compaction status may contribute to improving stability.
- 3) Two determinants are there for cellular uptake efficiency: RGD ligand and compact structure. Cellular uptake quantification, such as flow cytometry, should be performed to discriminate the contribution of these two determinants. 3 bs lantern is proper control.
- 4) Uncapped mRNA has immunogenic triphosphate at 5' end, which may influence the viability.
- 5) Does lantern structure influence RNA immunogenicity?
- 6) In vivo study simultaneously evaluates the contribution of two factors: RGD ligand and lantern shape. It would be helpful to have lantern-shaped mRNA without RGD and 3 bs RNA with RGD ligand for Smad4 expression assay.
- 7) l. 443, the sentence starting from Naoto should be referenced to identify the paper described.

Reviewer #2 (Remarks to the Author):

Thank you for the opportunity to review the article on: Lantern-shaped flexible RNA origami for Smad4 mRNA delivery and application in gene regulation therapy for colorectal cancer

Please see my comments below:

The work is important and could have broader implications other than in colorectal cancer given the importance of Smad4 across biology (e.g. key mediator in the TGF-beta superfamily, lung cancer, small bowel adenocarcinoma, juvenile polyposis, hereditary hemorrhagic telangiectasia and other pathologies).

The important results here are development of the technology for a) Smad4 mRNA delivery; b) construction of the nano-lantern; c) application of the technology in vitro and in vivo

The significance in the field is related to the importance of Smad4 in biology. It is a new technology so the work represents new knowledge and is important for potential future mechanistic work and development of therapeutics.

Relative to the data to support their claims please see my comments and concerns as I work through each section of the paper below.

Abstract: relatively well-written / may need to be re-structured in a revision

Introduction: too long and unfocused; I would strongly recommend a much more focused 3 paragraph introduction regarding the major problem at hand, the reason this technology is important and needed and the gaps that their science will fill.

Results:

Lines 118-121 of the results section - please clarify the meaning here - does this belong in the results section?? - Further, what does this statement really mean?

For construction and characterization of the mRNA nano-lantern: Fig 1c and Supp Figs 3&4  please clarify what the negative control is here

Relative to the use of the SW480 and SW620 cells - please clarify the source of these cells - this is crucial for the additional work in the paper as these cells are SMAD4 mutant and no SMAD4 should be expressed (see multiple examples of this in the literature):

PMID: 22115830; PMID: 30587545; PMID: 17308096; PMID: 33293694

They should provide references when they document the Smad4-null features of SW480/620 cells and any other cell lines used in future.

For me this is a critical part of the paper where what they do from this point forward is a bit difficult to follow when they are manipulating SW480 and SW620 cells that are not truly representative of the parental line (see Fig 2g).

If they are using contaminated cells then I would propose they repeat these experiments in ATCC SW480 and SW620 cells. It does not make me completely doubt all of their results, but it does make me worry that the tag line of the story is Smad4 and re-introduction of Smad4 into what should be Smad4-null cells. In addition, I would propose when they repeat these experiments that they use other cell lines known to be Smad4-null such as HT29 cells to broaden the biology and to move away from cell-type specific issues.

For Fig 2g - please clarify the negative control and why you would be getting a signal there - again I worry about the source of these lines.

For Figure 3 results:

- Please show better labeling to clarify what each fluorophore marks in the figure itself and the figure legend.

Figure 4 results:

- Please quantify the over-expression levels using nano-lantern vs lipofectamine (comparable and increased are not sufficient to describe the changes here)

- the authors indicate that Smad4 becomes nuclear and infer that it is functional, but show no data in this regard- the authors should demonstrate that translocation to the nucleus via their method leads to functional protein and expected downstream changes (show what happens to downstream targets when Smad4 becomes nuclear).

- again Fig. 4c and d - the negative control should not have a band - I worry that this indicates a contaminated line as this should be blank (e.g., no Smad4 band should be seen in Smad4-null lines)

Figure 5 results:

- this is mis-labeled as therapeutic effect - this is simply a demonstration of cell growth alterations, colony formation and cell migration changes - basic cancer biology - this has nothing to do with therapy. Further, there is nothing here related to or directly implicating a cause and effect. The authors should simply present the data and describe the results. It is highly encouraged for them to do this for each assay completed and not overstate the findings.

Also, they imply there is an anti-tumor effect here - they have shown in vitro data in Figure 5 and nothing related to anti-tumor biology (save this for the in vivo portion). Please remove this sentence.

Figure 6 results:

From my perspective, this segment needs to be reconfigured. Knocking down Smad4 in cells that should be Smad4-null does not make sense to this reviewer. A suggestion is that the authors simply add Smad4 to a cell line independent of the SW480/620 lines without Smad4 and then demonstrate MYC binding. It should be noted that the notion that Smad4 binds MYC is not a novel finding and has been demonstrated previously (PMID: 33293694). If the authors want to pursue this further to show the utility of the nano-lantern, I would support it, but it needs to be completed in a scenario where the cell lines are not a cause for doubting the validity of the entire experiment. Further, this section would further be strengthened by showing that by removing the nano-lantern that MYC repression is relieved.

Again - please note Figure 6c, f and g - negative control shows Smad4 detection in cell lines where a Smad4 band should not be detectable.

Fig. 7 results:

This work shows reduced tumorigenicity - this is a tumorigenesis experiment not a therapeutic experiment. If the authors were treating with a drug in parallel to the nano-lantern then I would support this phrasing, otherwise it is simply tumorigenicity.

I like the experimental design, but again this would be strengthened by using a truly Smad4-null line (see panel h).

Can the authors please clarify the color scheme in panel e - it is confusing as is especially given the color scheme used in panel d?

Can the authors please comment on why the H & E appears so different for the nano-lantern in panel f vs. mRNA?

Discussion: the discussion is brief and concise however it seems a unfocused and should rather focus on how their discoveries, limitations of the current work and next steps.

Materials and Methods:

The authors name no source for the SW480 and SW620 cell lines - further adding to my concern for the cells used in the experiments.

Lastly, maybe I missed it but I do not see a section discussing the antibodies used in this paper - this is important especially for the Smad4 blots.

Reviewer #3 (Remarks to the Author):

In this manuscript, the authors developed a novel mRNA delivery system using RNA origami. After optimization of the stability and cellular uptake, the 5 bs nano-lantern mRNA carrier induced obvious cellular uptake of mRNA and Smad4 protein expression in SW480 cells. Moreover, the author evaluated the in vitro and in vivo anti-tumor cells of the 5 bs nano-lantern mRNA carrier. To improve this mRNA delivery system, several concerns should be resolved.

(1) What are the endocytic pathways and endosome escape mechanisms of this nano-lantern mRNA carrier?

(2) For Figure 3b, the authors stated that the nano-lanterns escaped from the lysosome. Could the authors provide the quantified colocalization coefficient to demonstrate that?

(3) Whether the authors utilized chemical modified nucleotides, such as pseudouridine, in the IVT assay.

(4) The authors should validate the anti-tumor effect in one additional tumor model for increased confidence and versatility of this platform.

(5) In the discussion part, the authors stated that "However, its shortcomings are also obvious. The use of carriers brings new problems, such as biocompatibility. In the treatment of solid tumors, the targeting and response release of cargo also faces huge challenges." In fact, biocompatibility is dependent on the dose and biodegradability carriers. The currently approved mRNA nanoparticles including MC3, SM-102, and ALC-0315 are all biodegradable. Moreover, these carriers can be used for multiple administration routes, such as systemic administration. Additionally, these carriers have also been modified for targeting delivery (PMID: 30374059; PMID: 32251383). What are the advantages of the 5 bs nano-lantern mRNA carrier compared with the above delivery systems?

(6) What is the biodistribution profile of the 5 bs nano-lantern mRNA carrier post local injection?

Response to reviewers

Reviewer 1:

This manuscript addresses the use of RNA nano-structure for mRNA delivery without using mRNA delivery nanoparticles. mRNA was hybridized with circular RNA at 3 – 7 sites to prepare a lantern-shaped structure, and a targeting ligand was attached to circular RNA. The structuring improved mRNA nuclease stability, with the ligand enabling successful targeting. Without the use of delivery nanoparticles, their system is non-cytotoxic. The system was then used for delivering Smad4, a tumor suppressor gene, showing successful outcomes in vitro and in vivo. The concept is unique, and the experiments are well-designed. This manuscript is acceptable after addressing the following points.

We appreciate your constructive comments.

- 1) Why was the hybridization length determined to be 17 nt?

According to the previous report (PMID: 30669016, PMID: 31187576), the Satoshi Uchida *et al.* observed that the hybridization of mRNA with long oligonucleotides led to decreased mRNA translational activity and increased mRNA immunogenicity and that these undesirable responses after the formation of double stranded RNA (dsRNA) were avoided by limiting the hybridization length to 17 nt. However, the shortening of hybrid length tends to induce structural instability. To strike a balance between stability and translation efficiency, we limited the length of the complementary length of RNA to 17 nt in our work. We have also added the above description to the revised manuscript (Lines 117-119, Page 3).

- 2) It is interesting to find that 3 bs and 5 bs showed a large difference in compaction status. However, the difference in serum stability (Fig. 2a) is not large. Factors other than compaction status may contribute to improving stability.

Thanks for the comments. If only the two are compared without reference, it may indeed produce the confusion proposed by the reviewer. But, comparing individual mRNA, 3 bs, 5 bs and 7 bs together, it can be concluded from Fig. 1e that in terms of compaction there is a gradient: mRNA \ll 3 bs < 5 bs \approx 7 bs. And, for serum stability, the gradient is similar, i.e. mRNA \ll 3 bs < 5 bs \approx 7 bs (Fig. 2a & 2b), which is consistent with compaction. We have also added a short discussion in the revised manuscript to address this issue (Lines 161-162, Page 5).

- 3) Two determinants are there for cellular uptake efficiency: RGD ligand and compact structure. Cellular uptake quantification, such as flow cytometry, should be performed to discriminate the contribution of these two determinants. 3 bs lantern is proper control.

We are very grateful for your valuable comments. As suggested, we conducted experiments to investigate the two factors on cellular uptake efficiency by flow analysis. As shown in the Figure 1 below, as the binding sites increased (from 3 bs

to 7 bs), the cellular uptake efficiency increased, which indicates the more compact structure is conducive to the intake. In addition, the results showed RGD could help the cellular uptake of nanostructures, especially for 5 bs and 7 bs structures. While for 3 bs structure, the improvement of ingestion was not so obvious as 5 bs and 7 bs. We speculate that the larger and looser flexible structure of 3 bs has greater spatial steric hindrance, making its enhancement of ingestion limited. This result has also been added to the revised manuscript (Lines 174-176, Page 6, Revised Supplementary Fig. 6).

Figure 1. Intake profiles of FITC labeled nano-lantern (with 3, 5, and 7 bs) with or without RGD in SW480 cell.

- 4) Uncapped mRNA has immunogenic triphosphate at 5' end, which may influence the viability.

Thanks for the kind reminder. Only except in Figure 5a, we exploited the mature mRNA with capping on its 5' end all through the work to avoid potential immunogenicity. For the result shown in Figure 5a, the prone to degraded uncapped and untailed mRNA (Uu-mRNA) that cannot be translated was adopted to avoid the effects of Smad4 expression on cell viability. And the result shows uncapped mRNA as well as the nano-lantern structure doesn't influence the viability.

- 5) Does lantern structure influence RNA immunogenicity?

Thanks for the helpful comments. We investigated two proinflammatory factors interleukin-6 (IL-6) and interferon- β (IFN- β) levels in the peripheral blood of mice with orthotopic tumor models after injection with the nano-lantern. The results of the ELISA assay showed no significant immune response at 6 h after injection, indicating the low immunogenicity of nano-lantern (Figure 2 below, Figure 8d in revised MS). The result has been added to the revised manuscript (Lines 448-450, Page 16, Revised Figure 8d).

Figure 2. IL-6 and INF- β levels in peripheral blood of tumor-bearing mice at 6 h after intraperitoneal injection of different formulations.

- 6) In vivo study simultaneously evaluates the contribution of two factors: RGD ligand and lantern shape. It would be helpful to have lantern-shaped mRNA without RGD and 3 bs RNA with RGD ligand for Smad4 expression assay.

Thank you very much for your kind suggestion. As suggested, we have detected the protein level expression of 3, 5 and 7 bs groups treated with RGD and without RGD-modified nano lanterns (Figure 3 below, Supplementary Fig. S7 in revised MS). The western blot results showed that the transfection of RGD-modified nano lanterns can significantly increase their intracellular Smad4 expression compared to RGD-free groups. In the meanwhile, the Smad4 expression level of nano-lanterns with 5 bs was stronger than 7 bs and 3 bs, which is attributed to its more satisfying behavior of cellular uptake and intracellular release. The result has been added to the revised manuscript (Lines 182-183, Page 6, Revised Supplementary Fig. 7).

Figure 3. Western blot analysis of intracellular Smad4 expression in SW480 cells after transfection with 3 bs, 5 bs and 7 bs nano-lanterns with or without RGD.

- 7) l. 443, the sentence starting from Naoto should be referenced to identify the paper described.

As suggested, the relevant paper has been cited as Ref. 29 in our revised manuscript.

Finally, we would like to express our sincere gratitude. Thank you very much for your valuable comments on this work.

Reviewer 2:

Thank you for the opportunity to review the article on: Lantern-shaped flexible RNAorigami for Smad4 mRNA delivery and application in gene regulation therapy for

colorectal cancer.

Please see my comments below:

The work is important and could have broader implications other than in colorectal cancer given the importance of Smad4 across biology (e.g. key mediator in the TGF- β superfamily, lung cancer, small bowel adenocarcinoma, juvenile polyposis, hereditary hemorrhagic telangiectasia and other pathologies).

The important results here are development of the technology for a) Smad4 mRNA delivery; b) construction of the nano-lantern; c) application of the technology in vitro and in vivo.

The significance in the field is related to the importance of Smad4 in biology. It is a new technology so the work represents new knowledge and is important for potential future mechanistic work and development of therapeutics.

Relative to the data to support their claims please see my comments and concerns as I work through each section of the paper below.

We are very grateful for your excellent suggestion on our manuscript.

- 1) Abstract: relatively well-written / may need to be re-structured in a revision.
As suggested, we have revised this section (Lines 6-7, 11-14, Page 1).
- 2) Introduction: too long and unfocused; I would strongly recommend a much more focused 3 paragraph introduction regarding the major problem at hand, the reason this technology is important and needed and the gaps that their science will fill.
As suggested, we have revised the introduction part (Lines 20-26, 29-37, Page 1).
- 3) Results:
Lines 118-121 of the results section - please clarify the meaning here - does this belong in the results section? - Further, what does this statement really mean?
We apologize that we failed to make this sentence clear. The purpose of this sentence is to elaborate on the mechanism of Smad4 on its downstream genes in Scheme 1e and the application prospect of this protocol in colorectal cancer. To avoid making readers confused, we have revised this part (Lines 98-100, Page 3).
- 4) For construction and characterization of the mRNA nano-lantern: Fig 1c and Supp Figs 3&4  please clarify what the negative control is here.
Thanks for the helpful comments. Here, we intend to confirm the assembly of nano-lantern by agarose gel. The negative control is mRNA before assembly. After introducing the circular staples, mRNA can be folded into the nano-lantern structure. Therefore, the band of RNA nano-structure shifted (Fig 1c, lane 4) from individual mRNA (Fig 1c, lane 3), which indicated the successful construction of nano-lanterns. To make it clear, we added a relative description in the Results Section (Line 128, Page 4).
- 5) Relative to the use of the SW480 and SW620 cells - please clarify the source of these cells- this is crucial for the additional work in the paper as these cells are

SMAD4 mutant and no SMAD4 should be expressed (see multiple examples of this in the literature):PMID: 22115830; PMID: 30587545; PMID: 17308096; PMID: 33293694

They should provide references when they document the Smad4-null features of SW480/620 cells and any other cell lines used in future.

For me this is a critical part of the paper where what they do from this point forward is a bit difficult to follow when they are manipulating SW480 and SW620 cells that are not truly representative of the parental line (see Fig 2g).

If they are using contaminated cells then I would propose they repeat these experiments in ATCC SW480 and SW620 cells. It does not make me completely doubt all of their results, but it does make me worry that the tag line of the story is Smad4 and re-introduction of Smad4 into what should be Smad4-null cells. In addition, I would propose when they repeat these experiments that they use other cell lines known to be Smad4-null such as HT29 cells to broaden the biology and to move away from cell-type specific issues.

For Fig 2g - please clarify the negative control and why you would be getting a signal there- again I worry about the source of these lines.

Thank you very much for your valuable and constructive comments. As suggested, the SW480 and SW620 cell lines were authenticated, and the STR report showed that there was indeed cell contamination of HeLa cells (Figure 4 and 5 below). To ensure the reliability of the data, as suggested, we used the parental generation cells of SW480 and SW620 cell lines obtained from the American Type Culture Collection (ATCC) to re-conduct all the cell and animal experiments. The STR reports of parental SW480 and SW620 cell lines were supplied below (Figure 6 and 7 below). The western blot results of new cell lines showed almost invisible bands of Smad4 (Revised Fig. 2g, 4c, 4d, 6c, 6f, 6g, 7h, 8g; Revised Fig S7, S9, S10, and S11). We are sorry for not doing the cell identification earlier. Thank you again for your important reminder.

Figure 4. STR profiles of SW480 cell with HeLa contamination. The red arrows represent the peaks of HeLa cells.

Figure 5. STR profiles of SW620 cell with HeLa contamination. The red arrows represent the peaks of HeLa cells.

Figure 6. STR profiles of parental SW480 cell without contamination.

Figure 7. STR profiles of parental SW620 cell without contamination.

- 6) For Figure 3 results:
 - Please show better labeling to clarify what each fluorophore marks in the figure itself and the figure legend.

We have marked the fluorescence dye more clearly in the figure and the figure legend (Lines 250-252, Page 8, Revised Figure 3).
- 7) Figure 4 results:

Please quantify the over-expression levels using nano-lantern vs lipofectamine (comparable and increased are not sufficient to describe the changes here).

Thanks for the helpful comments. As suggested, we have quantified the over-expression levels using nano-lantern vs lipofectamine. And the relative description was added in the Results Section (Lines 275-278, Page 9, Revised Supplementary Fig. 11).
- 8) The authors indicate that Smad4 becomes nuclear and infer that it is functional, but show no data in this regard- the authors should demonstrate that translocation to the nucleus via their method leads to functional protein and expected downstream changes (show what happens to downstream targets when Smad4 becomes nuclear).

As kindly suggested by the reviewer, we have investigated the nuclear translocation of Smad4 after nano-lantern transfection by western blot. As shown in Figure 8 below, we could observe the protein expression level of Smad4 increased obviously in the cytoplasm and nucleus after transfected with Smad4 nano-lantern compared with empty nano-lantern. Quantitation of the nuclear and cytoplasmic western blot signals showed the increased nuclear-to-cytoplasmic ratio of Smad4 at 24 h post-transfection with Smad4 nano-lantern. In addition, the downstream

targets were then probed by western blot, and the results showed that nuclear translocation of Smad4 inhibited the expression of CXCL4, VEGFC and MYC. These data indicate that the Smad4 translocating to nuclear was mediated by nano-lantern, and functioned in inhibiting downstream CXCL4, VEGFC and MYC's expression. The result has been added to the revised manuscript (Line 283-288, Page 9; Line 371-375, Page 13, Revised Supplementary Fig. 12).

Figure 8. Nuclear translocation of Smad4 in SW480 cell lines after transfection with nano-lantern. (a) Western blot analysis of Smad4 expression in nucleus and cytoplasm and its downstream genes' expression. (b) Intensity statistical of the nucleus/cytoplasm ratio of Smad4 expression from panel a, n=3, Data are presented as the mean ± SEM. Statistical differences were assessed using one-way ANOVA with Bonferroni multiple comparisons test. *, p<0.05, **, p<0.01, ***, p<0.001.

- 9) Again Fig. 4c and d - the negative control should not have a band - I worry that this indicates a contaminated line as this should be blank (e.g., no Smad4 band should be seen in Smad4-null lines)

We appreciate your insightful concern. As mentioned in Q5, we have used the new parental cell lines to repeat these experiments.

- 10) Figure 5 results:

this is mis-labeled as therapeutic effect - this is simply a demonstration of cell growth alterations, colony formation and cell migration changes - basic cancer biology - this has nothing to do with therapy. Further, there is nothing here related to or directly implicating a cause and effect. The authors should simply present the data and describe the results. It is highly encouraged for them to do this for each assay completed and not overstate the findings.

Thanks for the valuable comments. As suggested, we have revised the description of the experiments' results to make it more appropriate (Lines 303, 315, 329, 332, Pages 10-12).

- 11) Also, they imply there is an anti-tumor effect here - they have shown in vitro data in Figure 5 and nothing related to anti-tumor biology (save this for the in vivo portion). Please remove this sentence.

Thanks for the kind reminder. We have revised this sentence in our revised manuscript (Line 329, Page 12).

- 12) Figure 6 results:

From my perspective, this segment needs to be reconfigured. Knocking down Smad4 in cells that should be Smad4-null does not make sense to this reviewer. A suggestion is that the authors simply add Smad4 to a cell line independent of the SW480/620 lines without Smad4 and then demonstrate MYC binding. It should be noted that the notion that Smad4 binds MYC is not a novel finding and has been demonstrated previously (PMID: 33293694). If the authors want to pursue this further to show the utility of the nano-lantern, I would support it, but it needs to be completed in a scenario where the cell lines are not a cause for doubting the validity of the entire experiment. Further, this section would further be strengthened by showing that by removing the nano-lantern that MYC repression is relieved.

Thanks for the suggestion. As mentioned in Q5, we have used new Smad4-null cells SW480 and SW620 to redo this experiment. The CHIP and western blot results showed the overexpression of Smad4 by nano-lantern suppressed the expression of downstream genes (Fig 6c-g).

In this study, we mainly aim to elaborate on the potential regulation application of nano-lantern in cells. Therefore, please allow us to verify the known downstream oncogenes of Smad4 here.

As suggested, we have probed the protein levels of downstream genes by removing the nano-lantern. The western blot results displayed the downstream genes regulated by Smad4 were recovered after removing the nano-lantern, which manifested the regulation function of Smad4 toward these genes (Figure 9 below, Supplementary Fig. 14 in MS). The result has been added to the revised manuscript (Lines 372-375, Page 13, Revised Supplementary Fig. 14).

Figure 9. Western blot analysis of Smad4, MYC, VEGFC and CXCL4 expression in SW480 cells after removing nano-lantern.

- 13) Again - please note Figure 6c, f and g - negative control shows Smad4 detection in cell lines where a Smad4 band should not be detectable.

We appreciate the reviewer's insightful concern. As mentioned in Q5, we have used new parental Smad4-null cells SW480 and SW620 to redid these experiments.

- 14) Fig. 7 results:

This work shows reduced tumorigenicity - this is a tumorigenesis experiment not a therapeutic experiment. If the authors were treating with a drug in parallel to the nano-lantern then I would support this phrasing, otherwise it is simply tumorigenicity.

Thanks for the valuable comments. We totally agree with you, and then we have revised this description (Lines 391-392, 414, 417, Pages 14-16).

- 15) I like the experimental design, but again this would be strengthened by using a truly Smad4-null line (see panel h).

Thanks for the valuable comments. We have used new parental Smad4-null cell SW480 and SW620 to redid relative experiments.

- 16) Can the authors please clarify the color scheme in panel e - it is confusing as is especially given the color scheme used in panel d?

Thanks for your kind reminder. We have revised this panel to make it clear.

- 17) Can the authors please comment on why the H&E appears so different for the nano-lantern in panel f vs. mRNA?

We are sorry to make you confused about these panels. The H&E panels aim to display the morphology of organ tissues and tumors. The light pink area represents the necrotic area of tumors. Actually, in each group, there are some light and dark

areas which represent the necrotic and normal. Since we redid this experiment, the new H&E pictures were displayed (Revised Figure 7g).

- 18) Discussion: the discussion is brief and concise however it seems a unfocused and should rather focus on how their discoveries, limitations of the current work and next steps.

Thanks for the valuable comments. We have revised the discussion part as suggested (Line 474-477, 480-481, 485-487, 488-490, 512-516, 519-521, Page 17-18).

- 19) Materials and Methods:

The authors name no source for the SW480 and SW620 cell lines - further adding to my concern for the cells used in the experiments.

Lastly, maybe I missed it but I do not see a section discussing the antibodies used in this paper - this is important especially for the Smad4 blots.

Thanks for your kind reminder. As suggested, the information on antibodies was attached in Supplementary Information as Table S2. The source of cells we used was added to the Materials and Methods Section.

Finally, we would like to express our sincere gratitude. Thank you very much for your valuable comments on this work.

Reviewer 3

In this manuscript, the authors developed a novel mRNA delivery system using RNA origami. After optimization of the stability and cellular uptake, the 5 bs nano-lantern mRNA carrier induced obvious cellular uptake of mRNA and Smad4 protein expression in SW480 cells. Moreover, the author evaluated the in vitro and in vivo anti-tumor cells of the 5 bs nano-lantern mRNA carrier. To improve this mRNA delivery system, several concernssould be resolved.

Many thanks for your comments.

- 1) What are the endocytic pathways and endosome escape mechanisms of this nano-lantern mRNA carrier?

Thanks a lot for your valuable comments. We explored the cellular uptake mechanism of mRNA nano-lantern in SW480 cells by using different endocytic inhibitors, including wortmannin (WTM, inhibitor of macropinocytosis), methyl-beta-cyclodextrin (MBCD, inhibitor of cholesterol-dependent endocytosis), chlorpromazine (CHL, inhibitor of clathrin-mediated endocytosis), and nystatin (NYS, inhibitor of lipid raft-caveolae endocytosis). As shown in Figure 10 below, after treatment with CHL and NYS, the fluorescence intensity in SW480 cells was significantly reduced, indicating that clathrin or lipid raft-caveolae endocytosis mediated the uptake process of nano-lantern. In addition, a small amount of nano-lantern relied on the cholesterol-dependent endocytosis in the cell, showing a slight decline in fluorescence intensity. The result has been added to the revised

manuscript (Lines 215-224, Pages 7, Revised Supplementary Fig. 8).

Figure S8. Investigation of cellular uptake mechanism. (a) Confocal images of SW480 cells after treated with nano-lanterns and different inhibitors. The inhibitors are as follows: chlorpromazine (CHL, inhibitor of clathrin-mediated endocytosis), amiloride (AMI, inhibitor of Na⁺/H⁺ pump related macropinocytosis), dynasore (DYN, inhibitor of dynamin), methyl-beta-cyclodextrin (MBCD, inhibitor of cholesterol-dependent endocytosis), and nystatin (NYS, inhibitor of lipid raft-caveolae endocytosis). Scale bar: 20 μ m. (b) Statistical results of the relative fluorescence intensities of Cy5 and FITC obtained from (a), n=3, Data are presented as the mean \pm SEM, Statistical differences were assessed using one-way ANOVA with Bonferroni multiple comparisons test. *, p<0.05, **, p<0.01, ***, p<0.001.

For the lysosome escape mechanism of nano-lantern, the previous reports revealed that DNA origami condensed by Mg²⁺ easily absorbed protons and released Mg²⁺ in acidic medium, disturbing osmotic pressure and inducing lysosomes to burst. Consequently, lysosomal escape of DNA origami was easily realized via proton sponge-like mechanism (PMID: 32725689; PMID: 30452233; PMID: 30566836; PMID: 31015729). Western blot results showed the Smad4 expression was markedly decreased in the presence of the proton pump inhibitor bafilomycin A1 (Baf A1), verifying this hypothesis (Figure 11 below, Supplementary Fig. 9 in MS). The description has been added to the revised manuscript (Lines 234-239, Page 7; Revised Supplementary Fig. 9).

Figure 11. Western blot analysis of Smad4 expression in SW480 cells after treatment with nano-lantern and Baf A1. The cells were pre-incubated for 30 min in serum-free medium containing Baf A1 (200 nM) inhibitors for intracellular proton-pump effects.

- 2) For Figure 3b, the authors stated that the nano-lanterns escaped from the lysosome. Could the authors provide the quantified colocalization coefficient to demonstrate that?

Thanks for your kind suggestion. We have presented the quantified colocalization coefficient of the confocal images (Revised Figure 3b).

- 3) Whether the authors utilized chemical modified nucleotides, such as pseudouridine, in the IVT assay.

Thank you very much for your valuable suggestion. In this experiment, we did not use pseudouridine which can increase the mRNA stability and decrease immune response. In our work, we modified the mRNA with 5' capped to increase its stability. And the mice results showed the nano-lanterns have low immunogenicity (Line 448-450, Page 16, Revised Figure 8d).

- 4) The authors should validate the anti-tumor effect in one additional tumor model for increased confidence and versatility of this platform.

Thanks for the kind suggestion. We have also investigated the tumor suppression of Smad4 mRNA nano-lantern in an orthotopic SW480-Luc tumors (Figure 12 below, Figure 8 and Supplementary Fig.15 in MS). The results have been added in the revised manuscript (Lines 432-471, Pages 16-17, Revised Figure 8 and Supplementary Fig.15).

Figure 12. Tumor suppression of Smad4 mRNA nano-lantern in an orthotopic model. (a) Systematic schedule of Smad4 mRNA nano-lantern administration in the orthotopic model of SW480-Luc cells. (b) Representative bioluminescence imaging of tumor burden and fluorescence imaging of Cy5-mRNA nano-lantern in tumor-bearing nude mice. (c) Ex vivo images of the xenograft tumors at the endpoint. (d) IL-6 and INF- β levels in peripheral blood of tumor bearing mice after 6 h post-intraperitoneal injection of different formulations. (e) IHC staining for Smad4 in tumor tissues. Scale bar: 100 μ m. (f) H&E staining of organs and tumor tissues in the tumor-bearing mice. Scale bar: 100 μ m. (g) Western blot analysis of Smad4, MYC, VEGFC, and CXCL5 expression in orthotopic tumors. Data are presented as the mean \pm SEM. Statistical differences were assessed using one-way ANOVA with Bonferroni multiple comparisons test. *, $p < 0.05$, **, $p < 0.01$, ***, $p < 0.001$.

- 5) In the discussion part, the authors stated that “However, its shortcomings are also obvious. The use of carriers brings new problems, such as biocompatibility. In the treatment of solid tumors, the targeting and response release of cargo also faces huge challenges.” In fact, biocompatibility is dependent on the dose and biodegradability carriers. The currently approved mRNA nanoparticles including MC3, SM-102, and ALC-0315 are all biodegradable. Moreover, these carriers can be used for multiple administration routes, such as systemic administration. Additionally, these carriers have also been modified for targeting delivery (PMID: 30374059; PMID: 32251383). What are the advantages of the 5 bs nano-lantern mRNA carrier compared with the above delivery systems?

Thanks a lot for your valuable comments. We have revised the discussion part to show the advantages, limitations and next steps (Line 474-477, 480-481, 485-490, 512-516, 519-521, Page 17-18). In comparison with the current strategies, nano-lanterns provide an accessible delivery method that avoids the use of chemical reagents and treatments. And in parallel, it also makes precise quantitative regulation of mRNA delivery possible through single-molecule nanolization.

- 6) What is the biodistribution profile of the 5 bs nano-lantern mRNA carrier post local injection?

Thanks for the helpful comments. As suggested, the biodistribution profile of 5 bs nano-lantern mRNA carrier post local injection was added in the revised manuscript (Lines 436-445, Page 16, Revised Figure 8b).

Finally, we would like to express our sincere gratitude. Thank you very much for your valuable comments on this work.

REVIEWERS' COMMENTS

Reviewer #1 (Remarks to the Author):

The manuscript has been properly revised with sufficient additional data and is thus now acceptable.

Reviewer #2 (Remarks to the Author):

The authors have responded to my queries in a reasonable manner - thank you.

Thank you for taking the time to address my concerns about the cell lines. I am reassured given the current results and quality control measures taken by the authors.

I have listed my additional comments below for consideration:

Abstract:

Proposed sentence for accuracy: lines 11-13:

The application of lantern-shaped flexible RNA in the context of the tumor suppressor gene, Smad4 in colorectal cancer models demonstrated promising potential for accurate manipulation of protein levels in in vitro and in vivo settings.

Introduction: no major issues

Results:

99: should be 'expression' NOT expressions

108: Figure 1 e legend  should state 'and potential use in studying growth suppression in colorectal cancer'

Figure 4: much better given the quality of the input with the correct SW480 and SW620 source cells. I would like to propose to the authors that showing the efficiency in one additional cell line would also further strengthen this methodology by demonstrating that the nano-lantern could be used in an equally efficient manner in HT29 or Colo205 cells (also SMAD4 deficient lines). They do NOT need to perform all of the additional experiments with these lines (in vitro and in vivo), but Figure 4 would be a good place to demonstrate this additional utility and to demonstrate this is a non- cell-type specific effect with another independent CRC line.

315-330:

Figure 5: please note that the features here should be mentioned specifically. I would suggest that the authors state they are investigating the 'tumor suppressive properties' of Smad4 including effects on proliferation, clonogenicity and migration. The authors should then simply state that they note a decrease in proliferation, clonogenicity and migration as supportive of the known tumor suppressor effects of Smad4 in CRC.

332: Figure 5 legend should read: Smad4 mRNA nano-lantern introduction correlates with reduced proliferation, clonogenicity and migration in vitro.

391: Use of tumor suppression is appropriate here as they are showing in vivo data

Figure 7: the photo in b seems a bit grainy - can they increase the sharpness of the image?

Figure 8c: similar issues - seems quite grainy in quality - can they improve the sharpness of the image

Also, no metastases in liver or lung were noted in their cecal injection model? The author should make mention of this - sorry if I missed it.

432: Please name this section: The tumor suppression of mRNA nano-lantern in an orthotopic cecal injection model (same for Figure 8).

Methods:

Agree with changes here

686-687: change to 'inoculated into the cecal wall. . . '

Reviewer #3 (Remarks to the Author):

Thanks for the authors' response. I have no further questions.

Response to reviewers

Reviewer #1:

The manuscript has been properly revised with sufficient additional data and is thus now acceptable.

Thank you very much!

Reviewer #2:

The authors have responded to my queries in a reasonable manner - thank you.

Thank you for taking the time to address my concerns about the cell lines. I am reassured given the current results and quality control measures taken by the authors.

I have listed my additional comments below for consideration:

We are very grateful for your constructive comments on our manuscript.

Abstract:

Proposed sentence for accuracy: lines 11-13:

The application of lantern-shaped flexible RNA in the context of the tumor suppressor gene, Smad4 in colorectal cancer models demonstrated promising potential for accurate manipulation of protein levels in in vitro and in vivo settings.

We appreciate your constructive comments and have replaced it with your sentence for accuracy (Lines 23-26, Page 1).

Introduction: no major issues.

Thanks!

Results:

99: should be 'expression' NOT expressions

Thanks. We have replaced expressions with 'expression' in our revised manuscript (Line 110, Page 3).

108: Figure 1e legend  should state 'and potential use in studying growth suppression in colorectal cancer'

We are very grateful for your valuable comments. We have added this sentence to Figure 1e legend.

Figure 4: much better given the quality of the input with the correct SW480 and SW620 source cells. I would like to propose to the authors that showing the efficiency in one additional cell line would also further strengthen this methodology by demonstrating that the nano-lantern could be used in an equally efficient manner in HT29 or Colo205 cells (also SMAD4 deficient lines). They do NOT need to perform all of the additional experiments with these lines (in vitro and in vivo), but Figure 4 would be a good place to demonstrate this additional utility and to demonstrate this is a non- cell-type specific effect with another independent CRC line.

Thanks for the helpful comments. We have investigated the Smad4 overexpression efficiency in HT29 cells by transfecting nano-lantern (Figure 5c and d). The Western blot results showed the non- cell-type specific effect of overexpressing Smad4 by our method (Lines 302-303, Page 9).

315-330: Figure 5: please note that the features here should be mentioned specifically. I would suggest that the authors state they are investigating the 'tumor suppressive properties' of Smad4 including effects on proliferation, clonogenicity and migration. The authors should then simply state that they note a decrease in proliferation, clonogenicity and migration as supportive of the known tumor suppressor effects of Smad4 in CRC.

Thank you very much for your kind suggestion. As suggested, we have added this description to our revised manuscript (Lines 345-346, 357-358, 360-361, Pages 11-12).

332: Figure 5 legend should read: Smad4 mRNA nano-lantern introduction correlates with reduced proliferation, clonogenicity and migration in vitro.

Thank you very much. As suggested, we have added this sentence to our revised Figure 6 legend.

391: Use of tumor suppression is appropriate here as they are showing in vivo data. Figure 7: the photo in b seems a bit grainy - can they increase the sharpness of the image? Figure 8c: similar issues - seems quite grainy in quality - can they improve the sharpness of the image. Also, no metastases in liver or lung were noted in their cecal injection model? The author should make mention of this - sorry if I missed it.

Thanks for the kind reminder. We have increased the sharpness of the images of Figure 8b and Figure 9c and added the description of metastases in liver or lung (Lines 490-491, Page 16).

432: Please name this section: The tumor suppression of mRNA nano-lantern in an orthotropic cecal injection model (same for Figure 8).

Thank you very much for your kind suggestion. As suggested, we have renamed this section (Lines 497-498, Page 17).

Methods: Agree with changes here

686-687: change to 'inoculated into the cecal wall. . . '

Thanks for the helpful comments. We have changed this description to 'inoculated into the cecal wall' in our revised manuscript (Lines 740, Page 23).

Reviewer #3:

Thanks for the authors' response. I have no further questions.

Thanks very much!

Finally, we would like to express our sincere gratitude. Thank you very much for your valuable comments on this work.